# LESS OR MORE FROM TEACHER: EXPLOITING TRILATERAL GEOMETRY FOR KNOWLEDGE DISTILLATION

**Chengming Hu**[1,2][*][†], **Haolun Wu**[1,2][*][†], **Xuan Li**[1,2], **Chen Ma**[3], **Xi Chen**[1], **Jun Yan**[4], **Boyu Wang**[5], **Xue Liu**[1,2]

{chengming.hu, haolun.wu, xuan.li2}@mail.mcgill.ca,
chenma@cityu.edu.hk, xi.chen11@mcgill.ca, jun.yan@concordia.ca,
bwang@csd.uwo.ca, xueliu@cs.mcgill.ca
[1]McGill University, [2]Mila - Quebec AI Institute, [3]City University of Hong Kong,
[4]Concordia University, [5]Western University

## ABSTRACT

Knowledge distillation aims to train a compact student network using soft supervision from a larger teacher network and hard supervision from ground truths. However, determining an optimal knowledge fusion ratio that balances these supervisory signals remains challenging. Prior methods generally resort to a constant or heuristic-based fusion ratio, which often falls short of a proper balance. In this study, we introduce a novel adaptive method for learning a sample-wise knowledge fusion ratio, exploiting both the correctness of teacher and student, as well as how well the student mimics the teacher on each sample. Our method naturally leads to the *intra-sample* trilateral geometric relations among the student prediction ($\mathcal{S}$), teacher prediction ($\mathcal{T}$), and ground truth ($\mathcal{G}$). To counterbalance the impact of outliers, we further extend to the *inter-sample* relations, incorporating the teacher's global average prediction ($\bar{\mathcal{T}}$) for samples within the same class. A simple neural network then learns the implicit mapping from the intra- and inter-sample relations to an adaptive, sample-wise knowledge fusion ratio in a bilevel-optimization manner. Our approach provides a simple, practical, and adaptable solution for knowledge distillation that can be employed across various architectures and model sizes. Extensive experiments demonstrate consistent improvements over other loss re-weighting methods on image classification, attack detection, and click-through rate prediction.

## 1 INTRODUCTION

Knowledge distillation (KD) (Hinton et al., 2015) is a widely used machine learning technique that aims to transfer the informative knowledge from a cumbersome model (i.e., teacher) to a lightweight model (i.e., student). The student is trained by both imitating the teacher's behavior and minimizing the difference between its own predictions and the ground truths. This is achieved by optimizing a convex combination of two losses: $\mathcal{L} = \alpha \mathcal{L}^{\text{KD}} + (1 - \alpha)\mathcal{L}^{\text{GT}}$, where $\alpha \in [0, 1]$ is the *knowledge fusion ratio* balancing the trade-off between the two different supervision signals.

Determining the knowledge fusion ratio $\alpha$ is critical for training. The most straightforward method is to pre-set an identical value for all training samples (Hinton et al., 2015; Huang et al., 2022; Clark et al., 2019; Romero et al., 2014; Park et al., 2019; Lassance et al., 2020). Other works, such as ANL-KD (Clark et al., 2019) and FitNet (Romero et al., 2014), gradually decrease $\alpha$ from 1 to 0 through an annealing factor. Recent studies (Lukasik et al., 2021; Zhou et al., 2021; Lu et al., 2021) imply that a uniform knowledge fusion ratio across all samples is sub-optimal and cannot well capture the nuanced dynamics of the knowledge transfer process, thus designing the knowledge fusion ratio in a more fine-grained manner. For instance, ADA-KD (Lukasik et al., 2021) assigns a higher $\alpha$ to a class if the teacher has a higher correctness on that class. WLS-KD (Zhou et al., 2021) takes

---

[*]Equal contribution with random order.

[†]To whom the correspondence should be addressed.

Our source code can be found here.

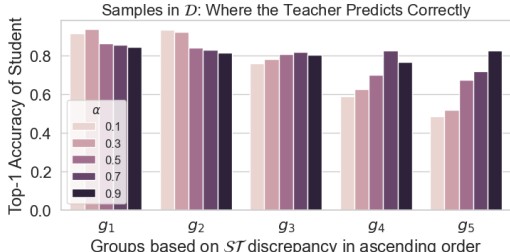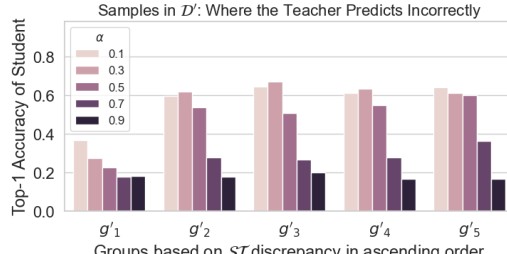

Figure 1: Motivation experiment on CIFAR-100 with a ResNet-34 teacher and a ResNet-18 student. The student is trained with varying knowledge fusion ratio values ($\alpha$). Data is first partitioned into $\mathcal{D}$ (where the teacher predicts correctly) and $\mathcal{D}'$ (incorrect predictions), and further categorized into five equalized groups based on the student-teacher prediction discrepancies ($\mathcal{ST}$), respectively. Our claim is that determining $\alpha$ greatly depends on $\mathcal{ST}$ and the correctness of teacher predictions.

both the teacher's and student's correctness into consideration, and $\alpha$ is increased if the teacher outperforms the student on a sample, otherwise decreased. RW-KD (Lu et al., 2021) analyzes the same information as WLS-KD yet employs a meta-learning method to learn the sample-wise $\alpha$.

However, existing methods largely ignore the discrepancy between the student's prediction ($\mathcal{S}$) and the teacher's prediction ($\mathcal{T}$), denoted as $\mathcal{ST}$, when determining $\alpha$. We argue that this oversight is significant, as making the student imitate the teacher lies at the heart of KD; thus intuitively, the $\mathcal{ST}$ discrepancy should offer valuable insights into balancing the two supervisory signals. Empirical results on CIFAR-100 (Krizhevsky, 2009) further verify our argument. The details of the motivation experiment are demonstrated at the end of this section. Derived from our observations, we draw the following insights:

- If the teacher predicts correctly, a higher $\mathcal{ST}$ discrepancy indicates the higher learning potential from the teacher, favoring a larger $\alpha$. A lower discrepancy indicates less potential to learn from the teacher and value in using the ground truth, thus a smaller $\alpha$ is preferred.

- If the teacher predicts incorrectly, knowledge from the teacher is misleading, and thus a smaller $\alpha$ is advisable.

- Regardless of the situation, determining a proper sample-wise value $\alpha$ relies on not only the teacher's or student's performances but also the value of $\mathcal{ST}$.

Consequently, our findings suggest that the $\mathcal{ST}$ discrepancy offers valuable insights for determining the knowledge fusion ratio $\alpha$. In light of the emphasized importance of student-ground truth ($\mathcal{SG}$) and teacher-ground truth ($\mathcal{TG}$) relations in existing studies (Zhou et al., 2021; Lu et al., 2021), we propose ***TGeo-KD***, which captures all three relations aforementioned and naturally leads to model the intra-sample **T**rilateral **Geo**metry among the signals from the student ($\mathcal{S}$), teacher ($\mathcal{T}$), and ground truth ($\mathcal{G}$). To enhance the model stability against outliers, we further incorporate the teacher's global average prediction for a given class as an additional reference, abbreviated as $\bar{\mathcal{T}}$, enriching the geometric relations at both intra- and inter-sample levels. Based on the insights from the motivation experiment, we also argue that learning the sample-wise $\alpha$ is quite involved and cannot be achieved by merely heuristic rules. To this end, we propose to learn the fusion ratio by a neural network (NN) and formulate KD as a bilevel objective that leverages the trilateral geometric information. As a result, the student is influenced by a tailored blend of knowledge from both the teacher and the ground truth. Our proposed *TGeo-KD*, an end-to-end solution, is versatile and proves superior to other re-weighting methods in various tasks, from image classification to click-through rate prediction. To summarize, the main contributions of our work are as follows:

- We introduce *TGeo-KD*, a novel method for learning sample-wise knowledge fusion ratios in KD. Leveraging the *trilateral geometry*, our method encapsulates the geometric relations among the signals from the student, teacher, and ground truth.

- We exploit the trilateral geometry at both intra-sample and inter-sample levels, mitigating the impact of outliers in training samples towards a more effective knowledge fusion.

- We conduct comprehensive experiments across diverse domains to demonstrate the consistent superiority over other loss re-weighting methods, as well as to highlight its versatility and adaptability across different architectures and model sizes.

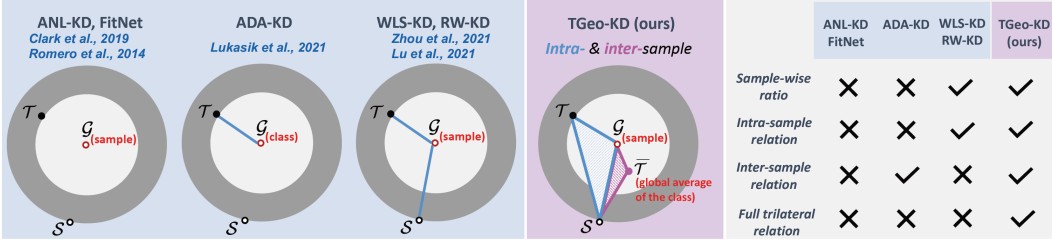

Figure 2: A comparison between prior works and our proposed TGeo-KD. The first two blocks show the relations captured in different methods for learning the knowledge fusion ratio from a geometric view on a sample. The third block shows crosscheck comparison on different method attributes. The details for representing each point and computing the geometric relation are demonstrated in Sec. 3.

**Details of Motivation Experiment.** We conduct our motivation experiment on CIFAR-100 (Krizhevsky, 2009). Specifically, we partition the dataset into two subsets, $\mathcal{D}$ and $\mathcal{D}'$, where $\mathcal{D}$ consists of samples on which the pre-trained teacher (ResNet-34) has correct predictions, whereas $\mathcal{D}'$ includes those with incorrect predictions. Initially, with $\alpha = 0.5$, we train a student model (ResNet-18) over 50 epochs to imbibe preliminary knowledge. We then compute the Euclidean distance between the student's and teacher's predicted class probabilities across all samples, designating this as the $\mathcal{ST}$ discrepancy. Based on ascending $\mathcal{ST}$ values, we further split $\mathcal{D}$ and $\mathcal{D}'$ into five equalized groups $g_1 \sim g_5$ and $g_1' \sim g_5'$, respectively. Subsequently, the student is further trained with varying $\alpha$ values adjusted from {0.1, 0.3, 0.5, 0.7, 0.9}, yielding five distinct student models. Upon evaluating these students across all five $g$ groups and five $g'$ groups, we obtain 25 bins for each subfigure in Fig. 1, which reveals that: (i) For samples in $\mathcal{D}$, students trained with smaller $\alpha$ values (i.e., 0.1, 0.3) outperformed on groups with lower $\mathcal{ST}$ discrepancies (i.e., $g_1$, $g_2$), whereas larger $\alpha$ values (i.e., 0.7, 0.9) were beneficial for groups with higher discrepancies (i.e., $g_4$, $g_5$). (ii) For samples in $\mathcal{D}'$, a smaller $\alpha$ (i.e., 0.1, 0.3) demonstrated the best performance on all $g'$ groups. Our observation shows a proper sample-wise value $\alpha$ relies on not only the student's or teacher's performances but also their discrepancy, which motivates the design of our proposed method for knowledge fusion learning.

## 2 PRELIMINARY: REVISITING KNOWLEDGE FUSION RATIO IN KD

The vanilla KD (Hinton et al., 2015) transfers knowledge from a pre-trained teacher network to a student by reducing discrepancies between their predictions and aligning with the ground truth. The student learns through two losses: $\mathcal{L}^{\text{KD}}$, the Kullback–Leibler (KL) divergence (Joyce, 2011) between student and teacher predictions, and $\mathcal{L}^{\text{GT}}$, the Cross-Entropy (CE) loss (Good, 1952) from the ground truth. Formally, denoting $\mathcal{D} = \{(\mathbf{x}_i, \mathbf{y}_i)\}_{i=1}^{N}$ as the data where $\mathbf{y}_i$ is the ground truth label represented as a one-hot vector for each sample, $C$ as the number of classes, $\mathbf{z}_i^s$ and $\mathbf{z}_i^t$ as the logits of the student and teacher, we formulate the two losses in a sample-wise manner as follows:

$$\mathcal{L}_i^{\text{KD}} = \tau^2 \text{KL}(\mathbf{z}_i^s, \mathbf{z}_i^t) = \tau^2 \sum_{j=1}^{C} \sigma_j(\mathbf{z}_i^t/\tau) \log \frac{\sigma_j(\mathbf{z}_i^t/\tau)}{\sigma_j(\mathbf{z}_i^s/\tau)}, \tag{1}$$

$$\mathcal{L}_i^{\text{GT}} = \text{CE}(\mathbf{z}_i^s, \mathbf{y}_i) = -\sum_{j=1}^{C} \mathbf{y}_{i,j} \log\left(\sigma_j(\mathbf{z}_i^s)\right), \tag{2}$$

where $\sigma$ is the softmax function and the temperature $\tau$ controls the softness of logits. Then the overall training objective aims to optimize the student network (parameterized by $\theta$) through a convex combination of the two losses with a sample-wise fusion ratio $\alpha_i$:

$$\mathcal{L} = \min_{\theta} \frac{1}{N} \sum_{i=1}^{N} \alpha_i \mathcal{L}_i^{\text{KD}} + (1 - \alpha_i)\mathcal{L}_i^{\text{GT}}. \tag{3}$$

We present a comparison of prior knowledge fusion methods alongside our work in Fig. 2, emphasizing both the geometric relations captured for learning $\alpha$ and their distinctive model attributes. Evidently, our proposed TGeo-KD overcomes the constraints observed in previous approaches, thus leading to enhanced performance.

# 3 TRILATERAL GEOMETRY GUIDED KNOWLEDGE FUSION

## 3.1 ADAPTIVE LEARNING FOR KNOWLEDGE FUSION RATIO

To address the limitation of prior works and employing the insights from the motivation experiment as depicted in Sec. 1, we propose to adaptively learn the knowledge fusion ratio based on trilateral geometry within $(\mathcal{S}, \mathcal{T}, \mathcal{G})$ triplet using a separate network. For simplifying the notation, we consistently denote $\mathcal{S} := \sigma(\mathbf{z}^s) \in \mathbb{R}^{N \times C}$ and $\mathcal{T} := \sigma(\mathbf{z}^t) \in \mathbb{R}^{N \times C}$ as the prediction probabilities of the student and teacher, and $\mathcal{G} := \mathbf{y} \in \mathbb{R}^{N \times C}$ as the ground truth (i.e., each row is an one-hot vector). Given a training sample $(\mathbf{x}_i, \mathbf{y}_i)$, the knowledge fusion ratio can be correspondingly modeled as $\alpha_i = f_\omega(\Delta_i)$, where $f_\omega$ is one NN parameterized by $\omega$. The final layer of $f_\omega$ employs a sigmoid activation, ensuring that $\alpha_i \in (0, 1)$. For brevity, we omit explicitly writing the sigmoid function. $\Delta_i$ represents the unique geometric relation among $\mathcal{S}_i$, $\mathcal{T}_i$, and $\mathcal{G}_i$.

Our ultimate goal is to find the optimal sample-wise ratios $\alpha_i = f_\omega(\Delta_i)$ that enable the student network parameterized by $\theta$ to generalize well on test data. This naturally implies a bilevel optimization problem (Franceschi et al., 2018) with $\omega$ as the outer level variable and $\theta$ as the inner loop variable:

$$\min_\omega \mathcal{J}_{\text{val}}^{\text{outer}}(\theta^*(\omega)) = \frac{1}{N_{\text{val}}} \sum_{i=1}^{N_{\text{val}}} \mathcal{L}_i^{\text{GT}}, \tag{4}$$

$$\text{s.t. } \theta^*(\omega) = \underset{\theta}{\operatorname{argmin}} \, \mathcal{J}_{\text{train}}^{\text{inner}}(\theta, \omega) := \frac{1}{N_{\text{train}}} \sum_{i=1}^{N_{\text{train}}} f_\omega(\Delta_i) \mathcal{L}_i^{\text{KD}} + \left(1 - f_\omega(\Delta_i)\right) \mathcal{L}_i^{\text{GT}}. \tag{5}$$

On the inner level, we aim to train a student network given a fixed $\omega$ by minimizing the combined loss. On the outer level, the loss function on the validation set serves as a proxy for the generalization error of $\omega$. The goal for TGeo-KD is to find $\omega$ to minimize the validation loss. Note that Eq. 4 is an implicit function of $\omega$ as $\theta^*$ depends on $\omega$.

## 3.2 EXPLOITING TRILATERAL GEOMETRY

For modeling the trilateral geometry of $\Delta_i$, we propose to capture both intra-sample and inter-sample geometric relations. The details are demonstrated as follows.

**Intra-sample relations.** Given the $i^{th}$ sample, to capture the trilateral geometry of the $(\mathcal{S}_i, \mathcal{T}_i, \mathcal{G}_i)$ triplet, denoting as $\Delta_i^{\mathcal{STG}}$, we capture its three edges as outlined below:

$$\mathbf{e}_i^{sg} := [\mathcal{G}_i - \mathcal{S}_i] \in \mathbb{R}^C, \mathbf{e}_i^{tg} := [\mathcal{G}_i - \mathcal{T}_i] \in \mathbb{R}^C, \mathbf{e}_i^{st} := [\mathcal{T}_i - \mathcal{S}_i] \in \mathbb{R}^C. \tag{6}$$

The three edges represent the student's correctness, the teacher's correctness, and the discrepancy between the student and teacher, respectively. Previous research (Zhou et al., 2021; Lu et al., 2021) has affirmed the efficacy of the first two edges in guiding the learning of $\alpha$, while the third concept is our original contribution. We finally represent $\Delta_i^{\mathcal{STG}}$ by also incorporating the exact three vertices $\mathcal{S}_i, \mathcal{T}_i, \mathcal{G}_i$, to capture the exact probability across all classes for incorporating more information:

$$\Delta_i^{\mathcal{STG}} := [\mathbf{e}_i^{sg} \oplus \mathbf{e}_i^{tg} \oplus \mathbf{e}_i^{st} \oplus \mathcal{S}_i \oplus \mathcal{T}_i \oplus \mathcal{G}_i], \tag{7}$$

where $\oplus$ is the concatenation operation.

**Inter-sample relations.** In addition to intra-sample relations, we argue that inter-sample relations are also essential for knowledge fusion learning, especially considering the impact of outliers in training samples. Out-of-distribution samples, which are significantly different from normal training data, commonly behave as outliers to challenge the generalization capability of a model (Lee et al., 2018; Wang et al., 2022). In KD, the teacher network may perform poorly on these outliers, occasionally even with high absolute values of confidence margin. Therefore, blindly using the teacher's prediction as the supervisory signal can result in the propagation of misleading knowledge, thereby disturbing the student's training process.

To address this issue, we introduce inter-sample geometric relations. For each sample, we associate it with an additional vertex $\overline{\mathcal{T}}_{c^i} \in \mathbb{R}^C$, representing the teacher's global average prediction for all

samples of the class ($c^i$) that sample $i$ belongs to. It is essential to understand that while each sample is linked to its respective class-specific vertex, samples within the same class refer to the same vertex $\bar{\mathcal{T}}_{c^i}$. Consequently, we incorporate an additional triplet, $(\mathcal{S}_i, \bar{\mathcal{T}}_{c^i}, \mathcal{G}_i)$, to encapsulate these inter-sample relations. This is achieved by a similar process as before, focusing on the three edges, as well as incorporating all the vertices as follows:

$$\Delta_i^{\mathcal{S}\bar{\mathcal{T}}\mathcal{G}} := [\mathbf{e}_i^{sg} \oplus \mathbf{e}_i^{\bar{t}g} \oplus \mathbf{e}_i^{s\bar{t}} \oplus \mathcal{S}_i \oplus \bar{\mathcal{T}}_{c^i} \oplus \mathcal{G}_i]. \tag{8}$$

As such, by introducing the teacher's average prediction at the inter-sample level, the exploitation of more supportive knowledge can be further facilitated to effectively guide the student training process, particularly in addressing outliers.

**Improved distillation with trilateral geometry.** Although we can fully explore the sample-wise trilateral geometry through intra- and inter-sample trilateral relations, it is still challenging to design an explicit formulation between these signals and a knowledge fusion ratio as depicted in Sec. 1. We thus use a simple network $f_\omega(\cdot)$ parameterized by $\omega$, to adaptively learn a flexible and sample-wise knowledge fusion ratio with the input of geometric relations. The information captured for each sample can be represented as follows:

$$\Delta_i := \Delta_i^{\mathcal{S}\mathcal{T}\mathcal{G}} \oplus \Delta_i^{\mathcal{S}\bar{\mathcal{T}}\mathcal{G}}, \tag{9}$$

$$:= [\mathbf{e}_i^{sg} \oplus \mathbf{e}_i^{tg} \oplus \mathbf{e}_i^{st} \oplus \mathbf{e}_i^{\bar{t}g} \oplus \mathbf{e}_i^{s\bar{t}} \oplus \mathcal{S}_i \oplus \mathcal{T}_i \oplus \bar{\mathcal{T}}_{c^i} \oplus \mathcal{G}_i], \tag{10}$$

where the redundant terms are removed for brevity. Through inputting $\Delta_i$ into $f_\omega(\cdot)$, the knowledge fusion ratio $\alpha_i$ can be adaptively learned, and $\omega$ is optimized with $\theta$ in an end-to-end way.

## 4 EXPERIMENTS

### 4.1 TASKS AND EXPERIMENT SETTINGS

**Tasks and datasets.** To demonstrate the broad applicability of our method, we conduct extensive experiments on **three different tasks**. Specifically, we use CIFAR-100 (Krizhevsky, 2009) and ImageNet (Deng et al., 2009) for **image classification** in computer vision, HIL (Pan et al., 2015) for **attack detection** in cyber-physical systems, and Criteo (Jean-Baptiste Tien, 2014) for **click-through rate (CTR) prediction** in recommender systems. Details of datasets and task selection are shown in Appendix A.1.

**Experiment settings.** In the experiment setup, the temperatures ($\tau$) are set as 4.0, 1.5, 1.5, and 2.0 on the four datasets, respectively. In the vanilla KD, the pre-set fusion ratios are 0.2, 0.3, 0.1, and 0.3, respectively. Considering that the original HIL (Pan et al., 2015) and Criteo (Jean-Baptiste Tien, 2014) are imbalanced, we conduct oversampling on the minority class of training set as the data pre-processing procedure, ensuring all classes have the equal number of samples in a balanced setting. We conduct the experiments on one NVIDIA RTX-3080 GPU and one RTX-3090 GPU. The detailed experiment settings can be found in Appendix A.2.

### 4.2 STUDENT CLASSIFICATION PERFORMANCE

**Results on CIFAR-100.** We evaluate our proposed TGeo-KD method against numerous established KD baselines, as illustrated in Table 1. To ascertain the significance of improvement, we conduct a statistical t-test across five repeated runs, with t-scores being calculated based on the top-1 classification accuracy of TGeo-KD and the baseline methods. All computed t-scores surpass the threshold value $t_{0.05,5}$, indicating the acceptance of the alternative hypothesis with a statistical significance level of 5.0%. This furnishes compelling evidence that TGeo-KD consistently demonstrates a marked enhancement in performance.

Notably, when the teacher (ResNet-56) and student (ResNet-20) models possess relatively similar architectures, ADA-KD (Lukasik et al., 2021) is the best baseline with a marginal improvement of 0.07% over the second-best baseline WLS-KD (Zhou et al., 2021). In comparison to ADA-KD (Lukasik et al., 2021), TGeo-KD illustrates a substantial advantage of 0.76%. As the architectural gap increases, like between the student ResNet-32 and the teacher ResNet-110, our method's performance advantage increases to 0.97%, compared to the best baseline. This performance gain

Table 1: Top-1 classification accuracy (%) on CIFAR-100. We re-implemented the methods denoted by * and calculated their average results (with standard deviation) over 5 repeated runs. For the remaining methods, we utilized the results provided or verified by the others (Tian et al., 2020; Zhou et al., 2021). The best performance is **bold**, while the second best is underlined.

| Method | Same architecture style | | | | | Different architecture styles | | |
|---|---|---|---|---|---|---|---|---|
| Teacher
Student | WRN-40-2
WRN-40-1 | ResNet-56
ResNet-20 | ResNet-110
ResNet-32 | ResNet-110
ResNet-20 | ResNet-32×4
ResNet-8×4 | ResNet-32×4
ShuffleNetV1 | ResNet-32×4
ShuffleNetV2 | WRN-40-2
ShuffleNetV1 |
| Teacher | 75.61 | 72.34 | 74.31 | 74.31 | 79.42 | 79.42 | 79.42 | 75.61 |
| Student | 71.98 | 69.06 | 71.14 | 69.06 | 72.50 | 70.50 | 71.82 | 70.50 |
| FitNet | 72.24 | 69.21 | 71.06 | 68.99 | 73.50 | 73.59 | 73.54 | 73.73 |
| AT | 72.77 | 70.55 | 72.31 | 70.22 | 73.44 | 71.73 | 72.73 | 73.32 |
| SP | 72.43 | 69.67 | 72.69 | 70.04 | 72.94 | 73.48 | 74.56 | 74.52 |
| CC | 72.21 | 69.63 | 71.48 | 69.48 | 72.97 | 71.14 | 71.29 | 71.38 |
| VID | 73.30 | 70.38 | 72.61 | 70.16 | 73.09 | 73.38 | 73.40 | 73.61 |
| RKD | 72.22 | 69.61 | 71.82 | 69.25 | 71.90 | 72.28 | 73.21 | 72.21 |
| PKT | 73.45 | 70.34 | 72.61 | 70.25 | 73.64 | 74.10 | 74.69 | 73.89 |
| AB | 72.38 | 69.47 | 70.98 | 69.53 | 73.17 | 73.55 | 74.31 | 73.34 |
| FT | 71.59 | 69.84 | 72.37 | 70.22 | 72.86 | 71.75 | 72.50 | 72.03 |
| NST | 72.24 | 69.60 | 71.96 | 69.53 | 73.30 | 74.12 | 74.68 | 74.89 |
| CRD | 74.14 | 71.16 | 73.48 | 71.46 | 75.51 | 75.11 | 75.65 | 76.05 |
| Vanilla KD | 73.54 | 70.66 | 73.08 | 70.67 | 73.33 | 74.07 | 74.45 | 74.83 |
| ANL-KD* | 72.81±0.25 | 72.13±0.18 | 72.50±0.21 | 72.28±0.21 | 75.07±0.26 | 72.58±0.23 | 73.11±0.14 | 75.27±0.32 |
| ADA-KD* | 74.67±0.19 | 72.22±0.21 | 73.19±0.12 | 72.29±0.27 | 75.78±0.34 | 71.45±0.16 | 72.20±0.24 | 75.49±0.28 |
| WLS-KD | 74.48 | 72.15 | 74.12 | 72.19 | 76.05 | 75.46 | 75.93 | 76.21 |
| RW-KD* | 73.92±0.22 | 70.33±0.26 | 71.78±0.15 | 71.24±0.16 | 74.86±0.29 | 70.45±0.25 | 70.69±0.17 | 74.15±0.29 |
| **TGeo-KD** | **75.43**±0.16 | **72.98**±0.14 | **75.09**±0.13 | **73.55**±0.20 | **77.27**±0.25 | **76.83**±0.17 | **76.89**±0.14 | **77.05**±0.23 |

Table 2: Top-1 and Top-5 classification accuracy on ImageNet. We re-implemented the methods denoted by * and used the author-provided or author-verified results for the others (Zhou et al., 2021).

| Teacher: ResNet-34 → Student: ResNet-18 | | | Teacher: ResNet-50 → Student: MobileNetV1 | | |
|---|---|---|---|---|---|
| Method | Top-1 ACC | Top-5 ACC | Method | Top-1 ACC | Top-5 ACC |
| Teacher | 73.31 | 91.42 | Teacher | 76.16 | 92.87 |
| Student | 69.75 | 89.07 | Student | 68.87 | 88.76 |
| AT | 71.03 | 90.04 | AT | 70.18 | 89.68 |
| NST | 70.29 | 89.53 | FT | 69.88 | 89.50 |
| FSP | 70.58 | 89.61 | AB | 68.89 | 88.71 |
| RKD | 70.40 | 89.78 | RKD | 68.50 | 88.32 |
| Overhaul | 71.03 | 90.15 | Overhaul | 71.33 | 90.33 |
| CRD | 71.17 | 90.13 | CRD | 69.07 | 88.94 |
| Vanilla KD | 70.67 | 90.04 | Vanilla KD | 70.49 | 89.92 |
| ANL-KD* | 71.83±0.22 | 90.21±0.26 | ANL-KD* | 70.40±0.15 | 89.25±0.22 |
| ADA-KD* | 71.96±0.17 | 90.45±0.21 | ADA-KD* | 71.08±0.24 | 90.17±0.16 |
| WLS-KD | 72.04 | 90.70 | WLS-KD | 71.52 | 90.34 |
| RW-KD* | 70.62±0.22 | 89.76±0.15 | RW-KD* | 70.15±0.16 | 89.40±0.19 |
| **TGeo-KD** | **72.89**±0.15 | **91.80**±0.04 | **TGeo-KD** | **72.46**±0.14 | **90.95**±0.17 |

further escalates to 1.26% when the student ResNet-20 and the teacher ResNet-110, underscoring the advantage of our TGeo-KD particularly when dealing with the increasing architectural disparities between the student and teacher. Moreover, our technique excels in hetero-architecture KD scenarios, wherein knowledge is distilled from either ResNet-32×4 or WRN-40-2 models into ShuffleNet. In these cases, our method consistently demonstrates performance enhancements of 1.37%, 0.96%, and 0.84%, respectively, compared to the strongest baseline WLS-KD (Zhou et al., 2021).

**Results on ImageNet.** To further demonstrate the effectiveness of our approach on larger datasets, we extend our experiments to ImageNet, adhering to the setup outlined by Zhou et al. (2021). As depicted in Table 2, TGeo-KD consistently outperforms all competing baselines. Notably, compared with the strongest baseline WLS-KD (Zhou et al., 2021), TGeo-KD exhibits the performance improvement of 1.10% when the teacher (ResNet-34) and student (ResNet-18) share the same architecture style. Similarly, in a hetero-architecture KD experiment with ResNet-50 and MobileNetV1 as teacher and student respectively, our method realizes an improvement of 0.94%.

Table 3: Result comparison on HIL and Criteo under Teacher (12-layer BERT) → Student (4-layer BERT). The best performance is **bold**, while the second best is underlined. "⇑" indicates the metric value the higher the better, while "⇓" indicates the lower the better. Our TGeo-KD demonstrates a statistical significance for $p \leq 0.01$ compared to the strongest baseline based on the paired t-test.

| Dataset | HIL | | | Criteo | | |
|---|---|---|---|---|---|---|
| Method | ACC (%) ⇑ | AUC (%) ⇑ | NLL ⇓ | ACC (%) ⇑ | AUC (%) ⇑ | NLL ⇓ |
| Teacher | 88.19 | 75.23 | 0.94 | 78.15 | 79.08 | 0.77 |
| Student | 87.64 | 67.58 | 1.02 | 69.43 | 69.02 | 1.79 |
| Vanilla KD | $87.55_{\pm0.56}$ | $69.52_{\pm0.70}$ | $1.00_{\pm0.04}$ | $71.08_{\pm0.48}$ | $69.42_{\pm0.60}$ | $1.51_{\pm0.05}$ |
| ANL-KD | $87.27_{\pm0.23}$ | $70.01_{\pm0.26}$ | $1.02_{\pm0.03}$ | $72.71_{\pm0.35}$ | $71.02_{\pm0.39}$ | $1.08_{\pm0.05}$ |
| ADA-KD | $90.15_{\pm0.34}$ | $70.02_{\pm0.21}$ | $0.99_{\pm0.02}$ | $72.15_{\pm0.33}$ | $71.01_{\pm0.35}$ | $1.15_{\pm0.04}$ |
| WLS-KD | $90.05_{\pm0.28}$ | $70.70_{\pm0.23}$ | $1.01_{\pm0.05}$ | $75.30_{\pm0.38}$ | $75.03_{\pm0.40}$ | $0.82_{\pm0.04}$ |
| RW-KD | $89.40_{\pm0.45}$ | $66.03_{\pm0.58}$ | $1.07_{\pm0.06}$ | $75.05_{\pm0.44}$ | $75.11_{\pm0.53}$ | $0.89_{\pm0.07}$ |
| **TGeo-KD** | **$92.39_{\pm0.49}$** | **$71.65_{\pm0.28}$** | **$0.94_{\pm0.03}$** | **$77.80_{\pm0.29}$** | **$77.00_{\pm0.32}$** | **$0.81_{\pm0.04}$** |

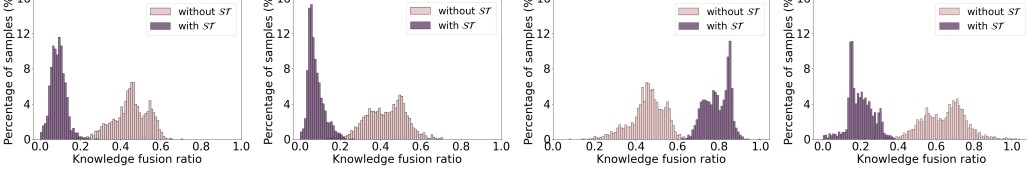

(a) Teacher ✗, large $\mathcal{ST}$  (b) Teacher ✗, small $\mathcal{ST}$  (c) Teacher ✔, large $\mathcal{ST}$  (d) Teacher ✔, small $\mathcal{ST}$

Figure 3: Knowledge fusion ratio distributions learned with (dark) and without (light) incorporating $\mathcal{ST}$ during learning $\alpha$. We first partition all samples into two subsets based on the teacher's correctness. In each subset, we sort the samples in descending order based on their $\mathcal{ST}$ values and select the top and bottom 20% as those with large and small discrepancies, respectively.

**Results on HIL and Criteo.**  To illustrate the broad applicability of our proposed TGeo-KD in diverse application scenarios, we also observe the similar superiority of our method on HIL for **attack detection** and Criteo for **CTR prediction**, as shown in Table 3. For instance, TGeo-KD not only relatively improves ACC and NLL over ADA-KD (Lukasik et al., 2021) by 2.48% and 5.05% on HIL, respectively, but also is better than the deeper teacher with the increasing ACC of 4.20%. Besides, WLS-KD (Zhou et al., 2021) and RW-KD (Lu et al., 2021) are the best methods among all the baselines on Criteo, approving the effectiveness of adopting sample-wise knowledge fusion ratio. More results on various network architectures are provided in Appendix A.4.

## 4.3  FUSION RATIO ANALYSIS WITH PREDICTION DISCREPANCY

To demonstrate the effectiveness of incorporating prediction discrepancy $\mathcal{ST}$ between the student and teacher on learning the knowledge fusion ratio $\alpha$, we follow the same settings as the motivation experiment in Sec. 1, for a comprehensive analysis. We first categorize training samples based on the correctness of teacher predictions and $\mathcal{ST}$ on these samples. We then compare the distributions of fusion ratio learned with and without the consideration of $\mathcal{ST}$, as depicted in Fig. 3. When the teacher predicts incorrectly, it may transfer misleading knowledge to the student, resulting in a decline in the student's performance. By incorporating $\mathcal{ST}$, our proposed TGeo-KD decreases the fusion ratio on $\mathcal{L}^{KD}$ when the discrepancy is either large or small (in Fig. 3(a) and Fig. 3(b)), which suggests that the student is encouraged to acquire more knowledge from the ground truths. In cases where the teacher predicts correctly, the fusion ratio typically ranges from 0.4 and 0.8 when not incorporating $\mathcal{ST}$. The fusion ratio is greater when incorporating $\mathcal{ST}$ in situations with a significant discrepancy between the student and teacher (in Fig. 3(c)). This suggests that the student is expected to emulate the teacher more closely, as the teacher possesses a greater potential for offering valuable knowledge. When the discrepancy is smaller, the student is encouraged to rely more on the ground truth, leading to a decline in the fusion ratio (in Fig. 3(d)). On the contrary, as illustrated in Table 4, the fusion ratio steadily increases without the incorporation of $\mathcal{ST}$ as the training progresses, yet an insufficient potential can be learned from the teacher.

Table 4: Comparison of average knowledge fusion ratios with and without $\mathcal{ST}$ during the training.

| Teacher predictions | ✗ | | | | ✗ | | | | ✔ | | | | ✔ | | | |
|---|---|---|---|---|---|---|---|---|---|---|---|---|---|---|---|---|
| Discrepancy | Large | | | | Small | | | | Large | | | | Small | | | |
| Epoch | 100 | 200 | 300 | 400 | 100 | 200 | 300 | 400 | 100 | 200 | 300 | 400 | 100 | 200 | 300 | 400 |
| Without $\mathcal{ST}$ | 0.51 | 0.46 | 0.43 | 0.42 | 0.48 | 0.42 | 0.40 | 0.39 | 0.48 | 0.55 | 0.57 | 0.57 | 0.52 | 0.59 | 0.62 | 0.64 |
| With $\mathcal{ST}$ | 0.25 | 0.19 | 0.14 | 0.12 | 0.19 | 0.13 | 0.10 | 0.08 | 0.59 | 0.67 | 0.71 | 0.73 | 0.42 | 0.33 | 0.29 | 0.27 |

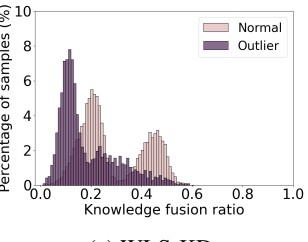
(a) WLS-KD

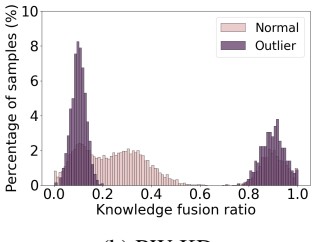
(b) RW-KD

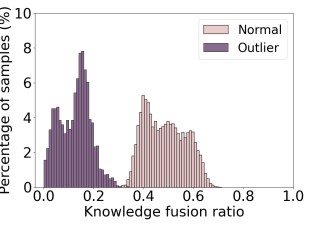
(c) TGeo-KD

Figure 4: Knowledge fusion ratio distributions on normal samples (light) and outliers (dark).

## 4.4 ANALYSIS ON NORMAL SAMPLES AND OUTLIERS

**Fusion ratio on normal samples and outliers.** To conduct analysis on fusion ratios between normal samples and outliers, we first create outliers by adding synthetic Gaussian noise as additional training samples following the setting outlied in the studies (Hendrycks and Gimpel, 2016; Liang et al., 2018; Vyas et al., 2018). In the context of Gaussian noise based outliers, each RGB value for every pixel is sampled from an independent and identically distributed Gaussian distribution with a mean of 0.5 and unit variance, and each pixel value is clipped to the range [0, 1]. As illustrated in Fig. 4, we compare the final fusion ratio distributions between sample-wise based baselines (i.e, WLS-KD (Zhou et al., 2021) and RW-KD (Lu et al., 2021)) and our TGeo-KD on CIFAR-100. Compared to the two baselines, TGeo-KD reports the final fusion ratios for normal samples within the range of 0.4 and 0.6 typically, which indicates that the student can be effectively guided with the valuable supervision signals from both the ground truth and the teacher on these normal samples. Furthermore, the teacher may make incorrect predictions on outliers, which provides misleading knowledge to the student. With the consideration of $\mathcal{ST}$ discrepancy and inter-sample relations, TGeo-KD reports the fusion ratios below 0.3 on outliers, which suggests that the student is expected to learn more informative knowledge from the ground truth, resulting in an increased fusion ratio on $\mathcal{L}^{\text{GT}}$. More comparisons about the effect of $\mathcal{ST}$ discrepancy and inter-sample relations on outliers can be found in Appendix A.4.

**Prediction discrepancy during the training and testing.** Fig. 5 shows the prediction discrepancies between the student and teacher on normal samples and outliers during training and testing. Compared to baselines, in Fig. 5(a), our proposed TGeo-KD achieves the smallest discrepancies (i.e., 0.19 during the training and 0.26 during the testing) on normal samples, indicating that the student can effectively mimic the teacher predictions through learning the knowledge distilled from the teacher. Furthermore, in Fig. 5(b), although ADA-KD (Lukasik et al., 2021) and RW-KD (Lu et al., 2021) (i.e, the baselines without inter-sample relations) report fewer discrepancies on outliers during the training, the teacher may report poor performance on these outliers and transfer misleading knowledge to the student. With the power of our inter-sample relations, TGeo-KD surpasses these aforementioned studies during the testing, which indicates that inter-sample relations can protect the student training process from being disrupted by low-quality knowledge from the teacher, especially on those outliers.

## 4.5 ABLATION STUDIES

**Effect of different relations captured.** TGeo-KD adaptively learns the knowledge fusion ratio by leveraging both intra- and inter-sample relations. To illustrate the effectiveness of each relation, we conduct experiments to train students under different combinations. As summarized in Table 6, both intra- and inter-sample relations yield better performance than the standalone student and vanilla KD models. Specifically, when we incorporate the $\mathcal{ST}$ relation, there is a notable increase in top-1 accuracy from 73.92% to 75.28%. This suggests the importance of accounting for the discrepancies between the student and teacher models. The performance improves to 76.83% when the inter-sample

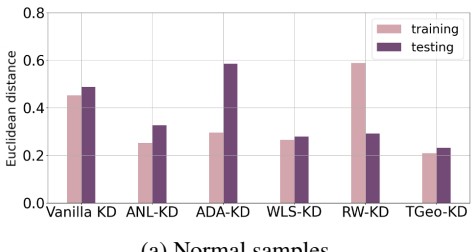 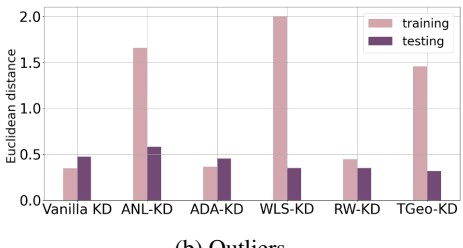

(a) Normal samples.         (b) Outliers.

Figure 5: Prediction discrepancies on normal samples and outliers during the training and testing.

Table 5: Top-1 classification accuracy (%) comparison among different modellings of $f_\omega(\cdot)$ through attention and MLP.

|  | CIFAR-100 | ImageNet |
|---|---|---|
| 1-head Atten. | 73.00 | 70.96 |
| 2-head Atten. | 73.63 | 71.72 |
| 3-head Atten. | 73.87 | 72.08 |
| 1-layer MLP | 72.69 | 71.74 |
| 2-layer MLP | **74.31** | **72.89** |
| 3-layer MLP | 73.96 | 72.52 |

Table 6: Ablation study on relations for learning $\alpha$ on CIFAR-100. The teacher and student are ResNet-32×4 and ShuffleNetV1.

|  | $\mathcal{SG}+\mathcal{TG}$ | $\Delta^{\mathcal{STG}}$ | $\Delta^{\mathcal{S\bar{T}G}}$ | ACC |
|---|---|---|---|---|
| Student | - | - | - | 70.50 |
| Vanilla KD | - | - | - | 74.07 |
| TGeo-KD | ✔ | - | - | 73.92 |
| TGeo-KD | ✔ | ✔ | - | 75.28 |
| TGeo-KD | ✔ | ✔ | ✔ | **76.83** |

relations are further captured. A deeper dive into how various representations of these intra- and inter-sample relations affect performance can be found in Appendix A.4.

**Comparison between attention mechanism and MLP.** We assess various options for modeling the knowledge fusion ratio generator, $f_\omega(\cdot)$, including attention mechanism as suggested in (Vaswani et al., 2017) and MLPs with different numbers of layers. For CIFAR-100, the teacher and student are ResNet-110 and ResNet-32. For ImageNet, the teacher and student are ResNet-34 and ResNet-18. Based on the results in Table 5, a 2-layer MLP is sufficient in capturing the valuable information embedded in the trilateral geometry, leading to superior performance across two datasets. Furthermore, MLP settings generally demonstrate a higher performance compared to attention mechanisms.

## 5 RELATED WORKS

**Knowledge balance in KD.** Knowledge distillation techniques have evolved from uniformly applying fusion ratios across samples (Hinton et al., 2015; Huang et al., 2022; Clark et al., 2019; Romero et al., 2014; Park et al., 2019; Lassance et al., 2020) to more refined strategies. Methods like ANL-KD (Clark et al., 2019) and FitNet (Romero et al., 2014) use annealing factors to adjust fusion ratios. Recent advancements include ADA-KD (Lukasik et al., 2021), which uses class-wise ratios, and WLS-KD (Zhou et al., 2021), which adjusts based on student's and teacher's performance. RW-KD (Lu et al., 2021) employs meta-learning for adaptive ratio learning. Yet, existing methods lack a consideration of the comprehensive trilateral relations among the signals from the student, teacher, and ground truth during the knowledge fusion learning.

**Sample relations in KD.** The exploitation of sample relations in knowledge distillation has been a key focus of numerous studies (Zagoruyko and Komodakis, 2017; Zhou et al., 2016; Park et al., 2019; Heo et al., 2019; Tian et al., 2020). For instance, AT (Zagoruyko and Komodakis, 2017; Hu et al., 2023) introduces attention transfer to transfer the attention maps from the teacher to the student, explicitly capturing the sample-level relation. CAMT (Zhou et al., 2016) extends this idea by incorporating both spatial and channel attention transfer to enhance the student's performance. Other studies have also emphasized relational reasoning and contrastive representation as key mechanisms for better understanding and improving knowledge distillation (Park et al., 2019; Tian et al., 2020). Despite these advancements in capturing sample relations, none of these studies target on the learning of the knowledge fusion ratio.

## 6 CONCLUSION

We propose an innovative approach named TGeo-KD for learning sample-wise knowledge fusion ratios during KD, which exploits the trilateral geometry among the signals from the student, teacher, and ground truth by modeling both intra- and inter-sample geometric relations. Across diverse domains, TGeo-KD outperforms other re-weighting methods consistently. It offers a simple yet adaptable distillation solution that fits various architectures and model sizes, thereby easing deployment complexities and resource limitations associated with deep neural networks. For the broader impact, TGeo-KD is effective across a variety of domains, including image classification, attack detection, and click-through rate prediction. Future research on TGeo-KD will address its focus on inter-sample relations within classes, exploring cross-class relationships to enhance performance.

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

# A  APPENDIX

## A.1  TASKS AND DATASETS

To demonstrate the effectiveness of our method TGeo-KD, we conduct extensive experiments across three diverse tasks, including **image classification**, **attack detection**, and **click-through rate (CTR) prediction**. The selection of the three tasks is based on their representations of diverse domains with different data natures, underlining the versatility of our proposed method. We use a classic and widely used dataset for each of the above tasks. This section provides the details for each dataset.

**CIFAR-100.**  The open-source benchmark dataset (Krizhevsky, 2009) is widely used for image classification, which comprises 60,000 color images of $32 \times 32$ pixels each. It is categorized into 100 classes, each containing 600 images. Each class is evenly split between the training and test sets, with 500 images for training and 100 for testing. This dataset includes a wide variety of objects, animals, and scenes, making it an ideal resource for developing and testing models and algorithms.

**ImageNet.**  ImageNet (Deng et al., 2009) is a large-scale dataset commonly used for image classification, containing millions of labeled images spanning thousands of categories, such as animals and scenes, etc. There are roughly 1.2 million training images, 50K validation images, and 150K testing images.

**HIL.**  The open-source benchmark dataset is generated from the HIL security testbed (Pan et al., 2015), the representative transmission network with rich operations in a realistic setting, which is widely used to develop proofs-of-concept for attack detection in industrial control systems. Specifically, this dataset includes 128 multi-sourced features, including 118 synchrophasor measurements and 12 descriptive system logs. The event scenarios can be categorized as three classes: normal operation (class 0), natural fault (class 1), and malicious attack (class 2). Our task is a three-class classification to identify the class of each event scenario.

As an imbalanced dataset, the normal and fault samples are randomly oversampled to balance the attack samples in the training set, while the validation and testing sets are not oversampled and remain imbalanced. To avoid dominating features, data normalization is essential to be performed to improve the uniformity of features by adjusting all feature values into the range of $[0, 1]$ (i.e., min-max normalization (Ioffe and Szegedy, 2015)).

**Criteo.**  Criteo is an online advertising dataset released by Criteo AI Lab . It contains feature values and click feedback for millions of display ads and serves as a benchmark for click-through rate (CTR) prediction. Each ad within the dataset is characterized by 40 attributes, providing a detailed description of the data. The first attribute acts as a label, indicating whether an ad was clicked (1) or not clicked (0), which also automatically forms the two classes in this dataset. The rest attributes consist of 13 integer features and 26 categorical features. To maintain anonymity, all feature values are hashed onto 32 bits. The dataset represents a full month of Criteo's traffic, offering extensive, real-world insights into online advertising dynamics.

The version we used in this paper is a well-known one that used in a competition on CTR prediction jointly hosted by Criteo and Kaggle in 2014 . Following prior works (Guo et al., 2017; Lyu et al., 2022), the continuous features are first normalized into the range of [0, 1]. Due to the imbalance of this dataset, we adopt the same strategy as HIL to conduct random oversampling to balance the two classes in this dataset.

## A.2  EXPERIMENT SETTINGS

Our method TGeo-KD can remain robust performance on all datasets, using different hyperparameter settings as shown in Table 7. This paper introduces three metrics for evaluating the student's generalization ability: *classification accuracy (ACC)*, *area under the ROC curve (AUC)*, and *negative log-likelihood (NLL)*. In order to avoid overfitting of the student, we conduct the early stopping

---

https://ailab.criteo.com/ressources/
https://www.kaggle.com/competitions/criteo-display-ad-challenge/

Table 7: Hyperparameter settings on different datasets.

| Dataset | batch size | optimizer | initial learning rate | weight decay | dim size |
|---------|-----------|-----------|----------------------|--------------|----------|
| CIFAR-100 | 128 | {Adam, SGD} | 1e-1 | 5e-4 | 128 |
| ImageNet | 256 | Adam | 1e-1 | 5e-4 | 128 |
| HIL | 256 | {Adam, SGD} | 1e-2 | {1e-2, 1e-1, 5e-1} | 32 |
| Criteo | 1024 | Adam | 1e-3 | {1e-5, 1e-3, 1e-2} | 64 |

Table 8: Classification accuracy (%) on DomainNet. All results are reported in terms of average classification accuracy (with standard deviation) over 5 repeated runs.

| Method | Clipart | Real | Sketch | Infograph | Painting | Quickdraw | Average |
|--------|---------|------|--------|-----------|----------|-----------|---------|
| Vanilla KD | $50.6_{\pm0.2}$ | $55.1_{\pm0.3}$ | $40.7_{\pm0.3}$ | $17.9_{\pm0.2}$ | $42.5_{\pm0.4}$ | $9.8_{\pm0.3}$ | $34.3_{\pm0.3}$ |
| ANL-KD | $52.5_{\pm0.4}$ | $57.0_{\pm0.3}$ | $41.9_{\pm0.3}$ | $19.6_{\pm0.2}$ | $44.6_{\pm0.3}$ | $11.3_{\pm0.3}$ | $37.8_{\pm0.3}$ |
| ADA-KD | $53.6_{\pm0.2}$ | $57.9_{\pm0.4}$ | $42.3_{\pm0.3}$ | $20.5_{\pm0.2}$ | $45.1_{\pm0.3}$ | $12.4_{\pm0.4}$ | $38.7_{\pm0.4}$ |
| WLS-KD | $53.1_{\pm0.4}$ | $57.7_{\pm0.3}$ | $42.8_{\pm0.2}$ | $21.1_{\pm0.1}$ | $45.3_{\pm0.2}$ | $12.0_{\pm0.2}$ | $38.5_{\pm0.3}$ |
| RW-KD | $51.3_{\pm0.3}$ | $56.2_{\pm0.3}$ | $41.2_{\pm0.2}$ | $18.8_{\pm0.3}$ | $43.7_{\pm0.2}$ | $10.6_{\pm0.3}$ | $37.1_{\pm0.2}$ |
| **TGeo-KD** | $\mathbf{55.2}_{\pm0.3}$ | $\mathbf{59.3}_{\pm0.4}$ | $\mathbf{44.6}_{\pm0.1}$ | $\mathbf{21.7}_{\pm0.3}$ | $\mathbf{46.9}_{\pm0.3}$ | $\mathbf{14.1}_{\pm0.3}$ | $\mathbf{40.2}_{\pm0.3}$ |

strategy when the ACC does not improve on the validation set within 10 successive iterations. Our computation resources include one NVIDIA RTX-3080 GPU and one RTX-3090 GPU.

### A.3 GENERALIZATION ABILITY

To demonstrate the generalization ability of our method on unseen data, we conduct additional experiments in the Out-of-Distribution Detection (OOD) task using DomainNet dataset (Peng et al., 2019). DomainNet (Peng et al., 2019) is a domain adaptation dataset with six domains (including Clipart, Real, Sketch, Infograph, Painting, and Quickdraw) and 0.6 million images distributed across 345 categories. In the experiment setup, we adopt a standard leave-one-domain-out manner, and train our student (teacher: ResNet-50 and student: ResNet-18) on the training set of the source domains and evaluate the trained student on all images of the held-out target domain. As depicted in Table 8, our TGeo-KD can outperform all the relevant KD-based baseline methods.

### A.4 MORE ABLATION STUDIES

**Student and teacher architectures.** In the main paper, we have shown the superiority of TGeo-KD across different architectures and model sizes of teachers and students on CIFAR-100 and ImageNet. Here, we provide more results on the other two datasets. As shown in Table 9, our proposed TGeo-KD consistently demonstrates strong performance across a range of architectural discrepancies between the teacher and student networks.

Table 9: Result comparison (with different gaps between students and teachers) on HIL and Criteo. "$S = i, T = j$" indicates the number of layers of the student and teacher network, respectively, which serve as a proxy for the size or the number of parameters or capacity of the network. Note that our baselines are built with "$S = 4, T = 12$" in the main paper.

| Dataset | HIL | | | Criteo | | |
|---------|-----|-----|-----|--------|-----|-----|
| Gap (in size) | ACC (%) ⇑ | AUC (%) ⇑ | NLL ⇓ | ACC (%) ⇑ | AUC (%) ⇑ | NLL ⇓ |
| $S = 4, T = 4$ | 89.33 | 69.48 | 1.02 | 74.37 | 74.09 | 0.97 |
| $S = 4, T = 8$ | 91.72 | 70.78 | 0.96 | 75.51 | 75.21 | 0.84 |
| $S = 4, T = 12$ | **92.39** | **71.65** | **0.94** | **77.80** | **77.00** | **0.81** |
| $S = 4, T = 16$ | 88.65 | 70.17 | 1.01 | 74.20 | 74.96 | 1.01 |
| $S = 4, T = 20$ | 87.92 | 69.58 | 1.03 | 73.28 | 73.27 | 1.08 |

Table 10: Sensitivity to temperature $\tau$. Top-1 classification accuracy (%) is reported.

| Dataset | $\tau=0.5$ | $\tau=1.0$ | $\tau=1.5$ | $\tau=2.0$ | $\tau=2.5$ | $\tau=3.0$ | $\tau=3.5$ | $\tau=4.0$ | $\tau=4.5$ | $\tau=5.0$ |
|---|---|---|---|---|---|---|---|---|---|---|
| CIFAR-100 | 69.80 | 71.51 | 71.65 | 72.30 | 72.07 | 72.15 | 72.09 | **72.98** | 72.53 | 71.22 |
| ImageNet | 71.05 | 71.76 | **72.89** | 72.28 | 72.53 | 71.74 | 72.01 | 71.63 | 71.82 | 71.44 |
| HIL | 91.04 | 92.12 | **92.39** | 91.70 | 91.87 | 91.65 | 92.05 | 91.31 | 91.52 | 91.10 |
| Criteo | 76.72 | 76.58 | 77.29 | **77.80** | 77.16 | 77.00 | 76.89 | 77.08 | 76.35 | 75.97 |

**Sensitivity to temperature $\tau$.** Table 10 compares the top-1 classification accuracy (%) of TGeo-KD using various temperatures (i.e., in the range of [0.5, 5.0]). It can be observed that our proposed TGeo-KD maintains stable performance using different temperatures $\tau$ (i.e., with the variances of 0.69, 0.26, 0.42, and 0.49), which further demonstrates the consistent performance of TGeo-KD across various temperature settings.

**Sensitivity to oversampling.** We employ oversampling when the dataset is imbalanced, which is a widely accepted practice in the research community, especially for classification tasks. To analyze the impact of oversampling on performance, we compare our TGeo-KD with five other baselines across varying imbalance ratios on the HIL and Criteo datasets. As shown in Table 11, our proposed TGeo-KD consistently outperforms all the baselines (e.g., the improved accuracy of 2.24% and 3.15% over the best baseline ADA-KD (Lukasik et al., 2021) with $\rho = 1$ and without oversampling on HIL dataset, respectively), which underscores the robustness of our TGeo-KD on imbalanced datasets.

Table 11: Top-1 classification accuracy (%) on HIL and Criteo with different imbalance ratios. The imbalance ratio $\rho$ is defined as the proportion between the sample sizes of the most prevalent class and the least prevalent class, where $\rho = 1$ represents a balanced setting. We calculate the average results (with standard deviation) over 5 repeated runs.

| Dataset | HIL | | | Criteo | | |
|---|---|---|---|---|---|---|
| Method | $\rho = 1$ | $\rho = 5$ | w/o | $\rho = 1$ | $\rho = 5$ | w/o |
| Vanilla KD | $87.55_{\pm 0.56}$ | $81.72_{\pm 0.66}$ | $72.07_{\pm 0.71}$ | $71.08_{\pm 0.48}$ | $67.32_{\pm 0.56}$ | $61.01_{\pm 0.39}$ |
| ANL-KD | $87.27_{\pm 0.23}$ | $81.33_{\pm 0.30}$ | $71.82_{\pm 0.29}$ | $72.71_{\pm 0.35}$ | $68.03_{\pm 0.38}$ | $62.18_{\pm 0.32}$ |
| ADA-KD | $90.15_{\pm 0.34}$ | $83.49_{\pm 0.22}$ | $75.23_{\pm 0.28}$ | $72.15_{\pm 0.33}$ | $67.74_{\pm 0.42}$ | $61.35_{\pm 0.46}$ |
| WLS-KD | $90.05_{\pm 0.28}$ | $82.83_{\pm 0.24}$ | $74.77_{\pm 0.27}$ | $75.30_{\pm 0.38}$ | $71.21_{\pm 0.55}$ | $64.52_{\pm 0.43}$ |
| RW-KD | $89.40_{\pm 0.45}$ | $82.22_{\pm 0.59}$ | $73.91_{\pm 0.58}$ | $75.05_{\pm 0.44}$ | $70.58_{\pm 0.37}$ | $63.90_{\pm 0.38}$ |
| **TGeo-KD** | $\mathbf{92.39}_{\pm 0.49}$ | $\mathbf{85.65}_{\pm 0.27}$ | $\mathbf{78.38}_{\pm 0.21}$ | $\mathbf{77.80}_{\pm 0.29}$ | $\mathbf{74.09}_{\pm 0.32}$ | $\mathbf{67.77}_{\pm 0.30}$ |

**Representations of Trilateral Geometry.** To fully exploit intra-sample and inter-sample relations, we consider multiple representations of trilateral geometry as follows:

- R1: $\mathcal{S}_i \oplus \mathcal{T}_i \oplus \bar{\mathcal{T}}_{c^i} \oplus \mathcal{G}_i$,
- R2: $\mathbf{e}_i^{sg} \oplus \mathbf{e}_i^{tg} \oplus \mathbf{e}_i^{st} \oplus \mathbf{e}_i^{\bar{t}g} \oplus \mathbf{e}_i^{s\bar{t}}$,
- R3: $\mathbf{e}_i^{sg} \oplus \mathbf{e}_i^{tg} \oplus \mathbf{e}_i^{st} \oplus \mathbf{e}_i^{\bar{t}g} \oplus \mathbf{e}_i^{s\bar{t}} \oplus \mathcal{S}_i \oplus \mathcal{T}_i \oplus \bar{\mathcal{T}}_{c^i} \oplus \mathcal{G}_i$,

where the explanations of notations can be found in the main paper. We conduct this ablation experiment on all the datasets. As shown in Table 12, our proposed TGeo-KD can achieve the best performance by capturing both the vertex-wise and edge-wise trilateral geometric information (i.e., R3) at the intra-sample and inter-sample levels.

**Effect of prediction discrepancy and inter-samples relations on outliers.** To analyze the impacts of prediction discrepancy ($\mathcal{S}\mathcal{T}$) and inter-sample relations ($\Delta^{\mathcal{S}\mathcal{T}\mathcal{G}}$) on outliers, we compare the Top-1 classification accuracy on outliers of CIFAR-100 when learning $\alpha$ based on different relations. As shown in Table 13, our proposed TGeo-KD achieves the best performance on outliers when considering both the discrepancy and inter-sample relations. Furthermore, compared to solely exploiting

Table 12: Top-1 classification accuracy (%) and AUC (%) comparison among different representations of trilateral geometry.

| Dataset | CIFAR-100 | | ImageNet | | HIL | | Criteo | |
|---|---|---|---|---|---|---|---|---|
| Representation | ACC | AUC | ACC | AUC | ACC | AUC | ACC | AUC |
| R1 | 70.32 | 66.24 | 69.87 | 64.52 | 87.52 | 69.82 | 73.87 | 73.39 |
| R2 | 68.44 | 64.10 | 71.28 | 65.91 | 87.14 | 70.26 | 74.21 | 74.01 |
| R3 | **72.98** | **68.57** | **72.89** | **67.14** | **92.39** | **71.65** | **77.80** | **77.00** |

Table 13: Comparison of Top-1 classification accuracy (%) on outliers among different relations. The teacher and student are ResNet-32×4 and ShuffleNetV1.

| | $\mathcal{SG}+\mathcal{TG}$ | $\mathcal{ST}$ | $\Delta^{\mathcal{S\bar{T}G}}$ | ACC |
|---|---|---|---|---|
| TGeo-KD | ✔ | - | - | 28.46 |
| TGeo-KD | ✔ | ✔ | - | 32.15 |
| TGeo-KD | ✔ | - | ✔ | 40.71 |
| TGeo-KD | ✔ | ✔ | ✔ | **45.89** |

the discrepancy, inter-sample relations make a better contribution to the improved performance on outliers.

Moreover, to analyze the impact of $\mathcal{ST}$ and $\Delta^{\mathcal{S\bar{T}G}}$ on the fusion ratios for incorrectly predicted samples, Table 14 reports the mean and standard deviation of the fusion ratios across these samples when learning fusion ratio with and without the consideration of $\mathcal{ST}$ and $\Delta^{\mathcal{S\bar{T}G}}$. When considering both $\mathcal{ST}$ and $\Delta^{\mathcal{S\bar{T}G}}$, our TGeo-KD demonstrates the fusion ratios below 0.3 for incorrectly predicted samples. This suggests that the student acquires more knowledge from the ground truth, leading to decreased fusion ratios on $\mathcal{L}^{\text{KD}}$. Furthermore, in comparison to solely exploiting $\Delta^{\mathcal{S\bar{T}G}}$ when learning fusion ratios, $\mathcal{ST}$ contributes more effectively to the decreased fusion ratios across incorrectly predicted samples.

Table 14: Final fusion ratios on incorrectly predicted samples with and without the consideration of prediction discrepancy ($\mathcal{ST}$) and inter-sample relations ($\Delta^{\mathcal{S\bar{T}G}}$).

| | $\mathcal{SG}+\mathcal{TG}$ | $\mathcal{ST}$ | $\Delta^{\mathcal{S\bar{T}G}}$ | Final fusion ratio |
|---|---|---|---|---|
| TGeo-KD | ✔ | - | - | 0.58±0.29 |
| TGeo-KD | ✔ | ✔ | - | 0.32±0.15 |
| TGeo-KD | ✔ | - | ✔ | 0.43±0.18 |
| TGeo-KD | ✔ | ✔ | ✔ | 0.12±0.07 |

**Computational cost.** We compare the computational cost of our proposed TGeo-KD with the baselines including vanilla KD (Hinton et al., 2015), ANL-KD (Clark et al., 2019), ADA-KD (Lukasik et al., 2021), and WLS-KD (Zhou et al., 2021). We run experiments on CIFAR-100, and report the mean training time (in seconds) on a batch of 128 images per iteration. The student network is ResNet-32. The experiment is conducted on one NVIDIA RTX-3090 GPU. As shown in Table 15, our method achieves better training efficiency.

## A.5 DISCUSSION

**Limitation.** Although our proposed TGeo-KD is a promising knowledge distillation approach, it could be limited by solely considering inter-sample relations within the same class, thereby neglecting relationships across different classes that could potentially enhance performance. This can be regarded as one future research direction of this work.

**Broader impact.** As for the broader impact, with its capability of dynamically learning a sample-wise knowledge fusion ratio across various model architectures, TGeo-KD holds the potential to

Table 15: Mean training time (in seconds) on a batch for CIFAR-100 per iteration.

| Vanilla KD | ANL-KD | ADA-KD | WLS-KD | **TGeo-KD (Ours)** |
|:---:|:---:|:---:|:---:|:---:|
| 0.095 | 0.095 | 0.101 | 0.097 | 0.097 |

significantly improve the performance of knowledge distillation across a variety of domains, including image classification, attack detection, and click-through rate prediction.

