# OpenReview forum: "Less or More From Teacher: Exploiting Trilateral Geometry For Knowledge Distillation"
_ICLR.cc/2024/Conference — ICLR 2024 poster_

### Official Review · Reviewer_aFDp · 2023-10-27

**Soundness:** 3 good
**Presentation:** 3 good
**Contribution:** 3 good
**Rating:** 8
**Confidence:** 4

**Summary:**

In this paper, the authors propose an adaptive method for learning a sample-wise knowledge fusion ratio. Specifically, the proposed method captures intra-sample and inter-sample trilateral geometric relations among the student prediction, teacher prediction, and ground truth. A simple network further learns the implicit mapping from the intra- and inter-sample relations to the fusion ratios in a bilevel-optimization manner. The proposed method is validated on three various tasks: image classification on CIFAR-100 and ImageNet, attack detection on HIL, and CTR prediction on Criteo.

**Strengths:**

1.	The paper is well motivated by clearly illustrating the valuable insights of incorporating additional ST prediction discrepancy for determining the fusion ratio.
2.	Although KD methods have been widely studied, the investigation of the knowledge learning potential on different samples has received less attention. With the learned sample-wise fusion ratio, this paper showcases that the student and teacher can provide specific knowledge learning potential for each sample based on the ST prediction discrepancy and the teacher’s correctness.
3.	This paper proposes a novel and interesting method to capture relations among the student prediction, teacher prediction, and ground truth by exploiting their trilateral geometry at both intra- and inter-sample levels.
4.	The experiments and analysis on three different tasks are comprehensive and demonstrate the effectiveness of the proposed method.
5.	The paper is well written and easy to follow.

**Weaknesses:**

The paper is overall readable and proposes an innovative idea. However, there are some weaknesses that the authors can improve.

1.	The experimental results show improved performance on the HIL and Criteo datasets. However, the paper does not clarify if this is due to the adopted sampling method. The oversampling of the minority class as a preprocessing step is mentioned, but its direct influence is not explored.

2.	In Section 4.3, this paper analyzes the effectiveness of incorporating ST by comparing the learned fusion ratio distributions across four scenarios. With ST, the proposed TGeo-KD reduces the fusion ratio on L_kd, thereby encouraging the student to acquire more knowledge from ground truths when the teacher predictions are incorrect on samples (e.g., outliers). However, more experiments should be conducted to investigate that the exploitation of inter-sample relations on these incorrectly predicted samples does not contribute significantly to decreased fusion ratios.

3.	The fusion ratio is learned through a neural network, which is unclear to explain how the learning process unfolds, such as which samples are assigned larger/smaller fusion ratios.

**Questions:**

1.	Why do the authors choose to model the relation in Euclidean space?
2.	Given that this paper proposes an adaptive learning method for sample-wise fusion ratio, does the proposed method demonstrate consistent performance across different numbers and categories of samples?
3.	Is there a specific reason to formulate this work into bilevel optimization? Will it lead to less efficient training? The motivation is note quite clear to me.

---

> ### Author Response · Authors · 2023-11-19
>
> We appreciate your time and feedback. We would also appreciate your agreement on the novelty, effectiveness, robustness, and adaptability of our method. Our reply to your questions is as follows.
>
> **Q1: Impact of oversampling on the performance.**
>
> **A1:** We employ oversampling when the dataset is imbalanced, which is a widely accepted practice in the research community, especially for classification tasks. To analyze the impact of oversampling on performance, we compare our TGeo-KD with five other baselines across varying imbalance ratios on the HIL and Criteo datasets. The imbalance ratio $\rho$ is defined as the proportion between the sample sizes of the most prevalent class and the least prevalent class, where $\rho=1$ represents a balanced setting. As shown in the tables below (Table 11 in the manuscript), we calculate the average classification accuracy (with standard deviation) over 5 repeated runs. TGeo-KD consistently outperforms all the baselines (e.g., the improved accuracy of 2.24% and 3.15% over the best baseline ADA-KD with $\rho=1$ and without oversampling on HIL dataset, respectively), which underscores the robustness of our TGeo-KD on imbalanced datasets. We have also provided the results in Appendix A.4 of the manuscript.
>
> *Table: Top-1 ACC on HIL.*
> |Method|$\rho=1$|$\rho=5$ |w/o oversampling|
> |:-:|:-:|:-:|:-:|
> |Vanilla KD|87.55±0.56|81.72±0.66|72.07±0.71|
> |ANL-KD|87.27±0.23|81.33±0.30|71.82±0.29|
> |ADA-KD|90.15±0.34|83.49±0.22|75.23±0.28|
> |WLS-KD|90.05±0.28|82.83±0.24|74.77±0.27|
> |RW-KD|89.40±0.45|82.22±0.59|73.91±0.58|
> |**TGeo-KD**|**92.39±0.49**|**85.65±0.27**|**78.38±0.21**|
>
> *Table: Top-1 ACC on Criteo.*
> |Method|$\rho=1$|$\rho=5$| w/o oversampling |
> |:-:|:-:|:-:|:-:|
> |Vanilla KD|71.08±0.48|67.32±0.56|61.01±0.39|
> |ANL-KD|72.71±0.35|68.03±0.38|62.18±0.32|
> |ADA-KD|72.15±0.33|67.74±0.42|61.35±0.46|
> |WLS-KD|75.30±0.38|71.21±0.55|64.52±0.43|
> |RW-KD|75.05±0.44|70.58±0.37|63.90±0.38|
> |**TGeo-KD**|**77.80±0.29**|**74.09±0.32**|**67.77±0.30**|
>
> **Q2: Impact of ST and inter-sample relations on the fusion ratio.**
>
> **A2:** To analyze the impact of prediction discrepancy $\mathcal{ST}$ and inter-sample relations $\Delta^{\mathcal{S\bar{T}G}}$ on the fusion ratios for incorrectly predicted samples, we have added the following table (Table 14 in the main manuscript), indicating the mean and standard deviation of the fusion ratios across these samples when learning fusion ratio with and without the consideration of $\mathcal{ST}$ and $\Delta^{\mathcal{S\bar{T}G}}$. When considering both $\mathcal{ST}$ and $\Delta^{\mathcal{S\bar{T}G}}$, our TGeo-KD demonstrates the fusion ratios below 0.3 for incorrectly predicted samples. This suggests that the student acquires more knowledge from the ground truth, leading to decreased fusion ratios on $\mathcal{L}^{KD}$. Furthermore, in comparison to solely exploiting $\Delta^{\mathcal{S\bar{T}G}}$ when learning fusion ratios, $\mathcal{ST}$ contributes more effectively to the decreased fusion ratios across incorrectly predicted samples. We have also provided the results in Appendix A.4 of the manuscript.
>
> |Method|$\mathcal{SG+TG}$|$\mathcal{ST}$|$\Delta^{\mathcal{S\bar{T}G}}$|Final fusion ratio on incorrectly predicted samples|
> |:-:|:-:|:-:|:-:|:-:|
> |TGeo-KD|✓|-|-|0.58±0.29|
> |TGeo-KD|✓|✓|-|0.32±0.15|
> |TGeo-KD|✓|-|✓|0.43±0.18|
> |TGeo-KD|✓|✓|✓|0.12±0.07|
>
> **Q3: Fusion ratio across different samples.**
>
> **A3:** Section 4.3 and Section 4.4 provide the analysis regarding the learning process of fusion ratios across normal samples and outliers. Specifically, Figure 4 shows the final fusion ratio distributions among sample-wise based baselines and our TGeo-KD on CIFAR-100. In comparison to the two baselines, our TGeo-KD increases the final fusion ratios for normal samples within the range of 0.4 and 0.6, which signifies the effective guidance of the student by valuable supervision signals from both the ground truth and the teacher. Additionally, as the teacher may make incorrect predictions on outliers, imparting misleading knowledge to the student, our TGeo-KD decreases the final fusion ratios below 0.3 for outliers. This indicates that the student is encouraged to learn more informative knowledge from the ground truth, resulting in decreased fusion ratios on $\mathcal{L}^{KD}$.
>
> Table 4 illustrates the average fusion ratios during the training. With the incorporation of $\mathcal{ST}$, our TGeo-KD consistently reduces the fusion ratios for incorrectly predicted samples as the training progresses. In cases where the teacher predicts accurately, the fusion ratio is increased in scenarios with a significant discrepancy between the student and teacher. This implies that the student is expected to emulate the teacher more closely, as the teacher possesses greater potential for offering valuable knowledge. On the contrary, when the discrepancy is smaller, the student is encouraged to rely more on the ground truth, resulting in a decline in the fusion ratio.

---

> ### Author Response · Authors · 2023-11-19
>
> **Q4: The choice of  Euclidean space.**
>
> **A4:** Our proposed method can model the sample-wise trilateral geometric relations in various spaces, including hyperbolic space and Euclidean space. Considering that Euclidean distance is a commonly used metric to model the relations among samples in knowledge distillation studies [6,7,8], we take Euclidean space as an illustrative example to represent the sample-wise trilateral geometry encompassing the student predictions, teacher predictions, and ground truth.
>
> **Q5: Generalization across different datasets.**
>
> **A5:** Our method showcases robust adaptability and generalization across a spectrum of datasets, each with its own unique characteristics in terms of sample categories and numbers.
> * ***CIFAR-100***: This dataset is a staple in the realm of knowledge distillation, encompassing 60,000 color images categorized into 100 distinct classes.  CIFAR-100 includes a wide variety of objects, animals, and scenes, standing as an excellent benchmark for the development and testing of models and algorithms. In comparison to the best baseline WLS-KD [5],  our method consistently exhibits performance enhancements of 1.37% and 1.26% across homo- and hetero-architectures knowledge distillation scenarios.
> * ***ImageNet***: ImageNet is a large-scale dataset commonly used for image classification, containing millions of labeled images spanning thousands of categories, such as animals and scenes, etc.  In comparison to the best baseline WLS-KD [5], our method consistently exhibits performance enhancements of 1.10% and 0.94% across homo- and hetero-architectures knowledge distillation scenarios.
> * ***HIL***: Primarily used for conceptualizing attack detection in industrial IoT systems, HIL emerges from a real-world transmission network, offering a tabular structure with 128 diverse features. Notably, it covers three event categories—normal operation (label 0), natural fault (label 1), and malicious attack (label 2) —with an inherent data imbalance among these classes (4,405, 18,309, and 55,663, respectively). In comparison to the best baseline ADA-KD [4], our method consistently exhibits performance enhancements of 2.24%.
> * ***Criteo***: This dataset is a benchmark for CTR prediction challenges. With a whopping 45 million user interactions, it features a mix of integer and categorical data attributes. It contains the majority (74.25%) of samples belonging to the "no-click" class (0) and a minority (25.75%) to the "click" class (1). In comparison to the best baseline WLS-KD [5], our method consistently exhibits performance enhancements of 2.50%.
>
> **Q6: Reason of formulating the problem into bilevel optimization.**
>
> **A6:** Our ultimate objective is to find the optimal sample-wise ratios $\alpha_{i}=f_{\omega}(\Delta_{i})$ that enable the student network parameterized by  to generalize well on test data. This naturally implies a bilevel optimization problem [1] with  as the outer level variable and  as the inner loop variable.
>
> In Appendix A.5 of the manuscript, the computational cost comparison of our proposed TGeo-KD with the baselines is provided in the table below. We run experiments on CIFAR-100 and report the mean training time (in second) on a batch of 128 images per iteration. The student network is ResNet-32. The experiment is conducted on one NVIDIA RTX-3090 GPU. As shown in the table below (Table 15 in the manuscript), TGeo-KD increases marginal training costs with significant accuracy improvement compared to other methods.
>
> *Table: Mean training time (in seconds) on a batch for CIFAR-100 per iteration.*
> ||Vanilla KD [2]|ANL-KD [3]|ADA-KD [4]|WLS-KD [5]|**TGeo-KD (ours)**|
> |:-:|:-:|:-:|:-:|:-:|:-:|
> |Training time (sec)| 0.095|0.095|0.101| 0.097|**0.097**|
> |ACC (%)|74.07|72.08|71.45|75.46|**76.83**|
>
> Ref:
> * [1] Franceschi, et al. Bilevel programming for hyperparameter optimization and meta-learning. ICML, 2018.
> * [2] Hinton et al. Distilling the knowledge in a neural network. NeurIPS, 2014.
> * [3] Clark et al. BAM! born-again multi-task networks for natural language understanding. ACL, 2019.
> * [4] Lukasik et al. Teacher’s pet: understanding and mitigating biases in distillation. TMLR, 2022.
> * [5] Zhou et al. Rethinking soft labels for knowledge distillation: A bias-variance tradeoff perspective. ICLR, 2021.
> * [6] Liu et al. Knowledge distillation via instance relationship graph. CVPR, 2019.
> * [7] Park et al. Relational knowledge distillation. ICCV, 2019.
> * [8] Romero et al. Fitnets: Hints for thin deep nets. ICLR, 2015.

---

> > ### Comment · Reviewer_aFDp · 2023-11-22
> >
> > Thank you very much for the authors' detailed response. After reading the rebuttal and other reviewer's comments, my concerns have been well addressed. Overall, I think this is a good work. Thus, I will increase my score correspondingly.

---

> > > ### Author Response · Authors · 2023-11-22
> > >
> > > Dear reviewer,
> > >
> > > Thank you again for your time and constructive comments. Your thoughtful evaluation and encouraging feedback significantly enhance the overall quality of our work.
> > >
> > > Best regards,
> > >
> > > Authors

---

### Official Review · Reviewer_iMjj · 2023-10-28

**Soundness:** 4 excellent
**Presentation:** 3 good
**Contribution:** 3 good
**Rating:** 8
**Confidence:** 5

**Summary:**

This paper focuses on data-wise adaptive knowledge fusion ratio in knowledge distillation. They find that the discrepancy of the predictions from teacher and student plays an important role, and they design a novel KD method based on this. Experiments show the effectiveness of their proposed method.

**Strengths:**

- They propose a novel KD method to exploit the discrepancy of the prediction logits from S and T to adaptively learn the knowledge fusion rate, offering an valuable view for KD.
- The experiments are solid and achieve good performance.
- More significantly, the proposed sample-wise operation introduces little additional training time, making this method useful in practice.

**Weaknesses:**

- How do the outliers hurt KD? As a key motivation, the corresponding empirical evidence seems lacking.
- In addition, the teacher model indeed may underperform on the outliers. According to previous discussion, directly decreasing the $\alpha$ also can solve this issue. Thus, why to introduce the inter-sample relations? This motivation needs to be further claified.
- Evaluating this method on real OOD datasets, like DomainNet, will be more convicing.

**Questions:**

- As far as I know, the computation of similarities of class probability distribution vectors often adopts the cosine similarity. Why do they use the Euclidean distance here? Are there some advantages over cosine similarity?
- As they state that Eq. 4 is an implicit function of ω as θ* depends on ω, how to perform alternately optimization?

---

> ### Author Response · Authors · 2023-11-19
>
> We appreciate your time and feedback. We would also appreciate your agreement on the novelty, effectiveness, robustness, and adaptability of our method. Our reply to your questions is as follows.
>
> **Q1: Empirical evidence of addressing outliers when learning fusion ratio.**
>
> **A1:** Given the substantial divergence between outliers and normal training data, the teacher network may perform poorly on these outliers, even with high absolute values of confidence margin. As illustrated in our motivation experiment presented in Figure 1 of the manuscript, the subset $D’$ includes samples on which the teacher makes inaccurate predictions. Notably, an increased fusion ratio  degrades the student’s performance across all $g’$ groups $(D’={g’_1, g’_2, g’_3, g’_4, g’_5})$. This observation underscores the risk of the teacher transferring misleading knowledge to the student when relying on the teacher’s prediction as the supervisory signal, ultimately leading to a decline in the student’s performance. Therefore, it is crucial to determine an appropriate sample-wise fusion ratio to effectively guide the student training process, particularly when addressing outliers.
>
> **Q2: Reason to introduce inter-sample relations.**
>
> **A2:** We argue that directly decreasing the same alpha value across all samples oversimplifies the varied impact of different outliers. Each outlier can uniquely influence the learning process, and a blanket reduction in alpha fails to address these nuances. We lack clarity on the extent of alpha value reduction needed for different samples without other guidance and signals.
>
> Our approach, which factors in inter-sample relations, aligns more closely with the principles of Knowledge Distillation, which is to understand the dynamics among the teacher, student, and ground truth. By examining these relationships, we can adjust the alpha value more precisely, tailoring the learning process to accommodate the diverse nature of outliers. Adopting this sophisticated, context-aware strategy is crucial for optimizing model performance, ensuring each outlier's impact is individually assessed and addressed, rather than resorting to a generalized, less effective solution.
>
> **Q3: Experiment on real OOD datasets.**
>
> **A3:** To better show the effectiveness of our method on Out-of-Distribution Detection (OOD) dataset, we conduct experiments on DomainNet. DomainNet [1] is a domain adaptation dataset with six domains (including Clipart, Real, Sketch, Infograph, Painting, and Quickdraw) and 0.6 million images distributed across 345 categories. In the experiment setup, we adopt a standard leave-one-domain-out manner, and train our student (teacher: ResNet-50 and student: ResNet-18) on the training set of the source domains and evaluate the trained student on all images of the held-out target domain. All results are reported in terms of average classification accuracy (with standard deviation) over 5 repeated runs. As reported in the table below (Table 8 in the manuscript), our TGeo-KD can outperform all the relevant KD-based baseline methods. We have also added the experimental results in Appendix A.3 of the manuscript.
>
> *Table: Classification accuracy (\%) on DomainNet.*
>
>
> ||Clipart|Real|Sketch|Infograph|Painting|Quickdraw|Avg|
> |:-:|:-:|:-:|:-:|:-:|:-:|:-:|:-:|
> |Vanilla KD|50.6±0.2|55.1±0.3|40.7±0.3|17.9±0.2|42.5±0.4|9.8±0.3|34.3±0.3|
> |ANL-KD|52.5±0.4|57.0±0.3|41.9±0.3|19.6±0.2|44.6±0.3|11.3±0.3|37.8±0.3|
> |ADA-KD|53.6±0.2|57.9±0.4|42.3±0.3|20.5±0.2|45.1±0.3|12.4±0.4|38.7±0.4|
> |WLS-KD|53.1±0.4|57.7±0.3|42.8±0.2|21.1±0.1|45.3±0.2|12.0±0.2|38.5±0.3|
> |RW-KD|51.3±0.3|56.2±0.3|41.2±0.2|18.8±0.3|43.7±0.2|10.6±0.3|37.1±0.2|
> |**TGeo-KD**|**55.2**±0.3|**59.3**±0.4|**44.6**±0.1|**21.7**±0.3|**46.9**±0.3|**14.1**±0.3|**40.2**±0.3|

---

> ### Author Response · Authors · 2023-11-19
>
> **Q4: The choice of  Euclidean space.**
>
> **A4:** Our proposed method can model the sample-wise trilateral geometric relations in various spaces, including Euclidean space and other measures. Considering that Euclidean distance is a commonly used metric to model the relations among samples in knowledge distillation studies [2,3,4], we take Euclidean space as an illustrative example to represent the sample-wise trilateral geometry encompassing the student predictions, teacher predictions, and ground truth. Our approach certainly accommodates the use of cosine similarity. However, we have not extensively explored this option, as it falls outside the central focus of our paper.
>
> **Q5: Iterative optimization procedure.**
>
> **A5:** To conduct the optimization for the bilevel optimization problem in Eq. 4, we speed up the training by using an approximation that has been validated in [5,6,7]. The key idea is to define a proxy function to link $\omega$ to the outer objective as follows: $\tilde{\theta}(\omega) := \theta - \alpha \nabla_{\theta}\mathcal{J}^{\text{inner}}(\theta, \omega)$, where $\alpha$ is the learning rate.
>
> Then the iterative optimization process from step $t$ to step $t+1$ can be illustrated as follows:
> * $\theta$ update: $\theta^{t+1} \leftarrow OPT_{\theta}\Big(\theta^t, \nabla_{\theta^t}\mathcal{J}^{\text{inner}}(\theta^t, \omega^t)\Big)$
> * Proxy: $\tilde{\theta}^{t+1}(\omega^t) := \theta^t - \alpha \nabla_{\theta^t}\mathcal{J}^{\text{inner}}(\theta^t, \omega^t)$
> * $\omega$ update: $\omega^{t+1} \leftarrow OPT_{\omega}\Big(\omega^t, \nabla_{\omega^t}\mathcal{J}^{\text{outer}}(\tilde{\theta}^{t+1}(\omega^t))\Big)$
>
> Ref:
> * [1] Peng et al. “Moment matching for multi-source domain adaptation.” ICCV, 2019.
> * [2] Liu et al. "Knowledge distillation via instance relationship graph." CVPR, 2019.
> * [3] Park et al. "Relational knowledge distillation." ICCV, 2019.
> * [4] Romero et al. “Fitnets: Hints for thin deep nets.” ICLR, 2015
> * [5] Liu et al. "Darts: Differentiable Architecture Search." ICLR, 2019.
> * [6] Chen et al. "λOpt: Learn to Regularize Recommender Models in Finer Levels." KDD, 2019.
> * [7] Ma et al. "Probabilistic Metric Learning with Adaptive Margin for Top-K Recommendation." KDD, 2020.

---

> > ### Comment · Reviewer_iMjj · 2023-11-22
> >
> > Thank you very much for the your detailed clarification. I have gone through your responses and other reviewer's comments, my concerns have been well addressed. Overall, I am willing to support this paper and keep my current score.

---

> ### Author Response · Authors · 2023-11-22
>
> Dear reviewer,
>
> Thank you again for your time and constructive comments. Your thoughtful evaluation and encouraging feedback significantly enhance the overall quality of our work.
>
> Best regards,
>
> Authors

---

### Official Review · Reviewer_UyVS · 2023-10-28

**Soundness:** 3 good
**Presentation:** 4 excellent
**Contribution:** 2 fair
**Rating:** 5
**Confidence:** 4

**Summary:**

This paper studies a less explored topic in Knowledge Distillation that of finding the optimal parameter/weight alpha for combining the distillation loss with the standard cross entropy loss when training the student. The authors first motivate the impact of appropriately learning the weight parameter with a well-designed experiment and then they propose a practical method to estimate it using a small network that uses as input features geometric features computed from the teacher's and student's logit and the ground truth one-hot vectors. They show good results on a variety of standard KD benchmarks

**Strengths:**

1. Paper is well-written and motivating example makes sense
2. Problem tackled is not well-studied and the solution proposed is simple and effective
3. Results carried out on standard benchmarks show that the method is effective on these benchmarks

**Weaknesses:**

1. My main concern is related to the impact of the proposed solution. Like most distillation papers conclusions are drawn based on CIFAR-100 experiments using same network architecture. Claims made by the authors could substantiate better if they are shown to hold on bigger datasets like ImageNet.
2. Tasks other than classification should be considered (e.g. detection). Experiments on HIL and Criteo datasets are of low impact
3. Important references/comparison with SOTA is insufficient. Authors should have included comparisons with Decoupled Knowledge Distillation, CVPR'22 and KNOWLEDGE DISTILLATION VIA SOFTMAX REGRESSION REPRESENTATION LEARNING, ICLR'21. From my preliminary checking, the improvements of the proposed method are not always so significant if the aforementioned methods are considered.

**Questions:**

Inter sample relations are used to model outliers. But how do you define outliers in the datasets you're using for your experiments (i.e. imagenet and cifar)?

---

> ### Author Response · Authors · 2023-11-19
>
> We appreciate your time and feedback. We would also appreciate your agreement on the novelty, effectiveness, robustness, and adaptability of our method. Our reply to your questions is as follows.
>
> **Q1: Experiment on ImageNet for higher impact.**
>
> **A1:** In Section 4.2, we conduct the experiments on ImageNet to further demonstrate the effectiveness of our approach on larger datasets. As depicted in the table below (Table 2 in the manuscript), TGeo-KD consistently outperforms all competing baselines. Compared with the best baseline WLS-KD [1], TGeo-KD exhibits the performance improvement of 1.10% when the teacher (ResNet-34) and student (ResNet-18) share the same architecture style. In a hetero-architecture KD experiment with ResNet-50 and MobileNetV1 as teacher and student respectively, our method realizes an improvement of 0.94%.
>
> *Table: Top-1 and Top-5 classification accuracy on ImageNet. We re-implemented the methods denoted by \* and used the author-provided or author-verified results for the others.*
>
> *Teacher: ResNet-34 → Student: ResNet-18*
>
> | Method| Top-1 ACC| Top-5 ACC|
> |:-:|:-:|:-:|
> |Teacher|73.31|91.42|
> |Student|69.75|89.07|
> |AT|71.03|90.04|
> |NST|70.29|89.53|
> |FSP|70.58|89.61|
> |RKD|70.40|89.78|
> |Overhaul|71.03|90.15|
> |CRD|71.17|90.13|
> |Vanilla KD|70.67|90.04|
> |ANL-KD\*|71.83 ±0.22|90.21 ±0.26|
> |ADA-KD\*|71.96 ±0.17|90.45 ±0.21|
> |WLS-KD| 72.04*|*90.70*|
> |RW-KD\*|70.62 ±0.22|89.76 ±0.15|
> |**TGeo-KD**|**72.89 ±0.15**|**91.80 ±0.04**|
>
> *Teacher: ResNet-50 → Student: MobileNetV1*
>
> |Method|Top-1 ACC|Top-5 ACC|
> |:-:|:-:|:-:|
> |Teacher|76.16|92.87|
> |Student|68.87|88.76|
> |AT|70.18|89.68|
> |FT|69.88|89.50|
> |AB|68.89|88.71|
> |RKD|68.50|88.32|
> |Overhaul|71.33|90.33|
> |CRD|69.07|88.94|
> |Vanilla KD|70.49|89.92|
> |ANL-KD\*|70.40±0.15|89.25±0.22|
> |ADA-KD\*|71.08±0.24|90.17±0.16|
> |WLS-KD|*71.52*|*90.34*|
> |RW-KD\*|70.15±0.16|89.40 ±0.19|
> |**TGeo-KD**|**72.46±0.14**|**90.95 ±0.17**|
>
> **Q2: Other tasks and impact.**
>
> **A2:** While existing knowledge distillation works have demonstrated success in image classification tasks, their applications in real-world industrial systems, including cyber-physical systems and recommender systems, still remain limited. Hence,  to illustrate the broad applicability of our approach across diverse application scenarios, we conduct extensive experiments spanning three tasks: CIFAR-100 and ImageNet for image classification, HIL for attack detection in cyber-physical systems, and Criteo for click-through prediction in recommender systems.
>
> Moreover, the selection of three tasks considers the representation of diverse domains characterized by different data natures. CIFAR-100 and ImageNet serve as widely recognized benchmarks for image-based knowledge distillation. HIL, as the representative of time series data, captures the dynamics of real-world industrial systems, thus presenting unique challenges different from image datasets [1-3]. Criteo is vital for its reflection of human behavior through its vast collection of feature values and click feedback for display ads, offering a distinct challenge relative to the others [4-6].
>
> In addition to the experiments included in the current manuscript, we conduct additional experiments on DomainNet dataset [7], to demonstrate the generalization ability of our method in the Out-of-Distribution Detection (OOD) task. DomainNet [7] is a domain adaptation dataset with six domains (including Clipart, Real, Sketch, Infograph, Painting, and Quickdraw) and 0.6 million images distributed across 345 categories. In the experiment setup, we adopt a standard leave-one-domain-out manner, and train our student (teacher: ResNet-50 and student: ResNet-18) on the training set of the source domains and evaluate the trained student on all images of the held-out target domain. All results are reported in terms of average classification accuracy (with standard deviation) over 5 repeated runs. As reported in the table below (Table 8 in the manuscript), our TGeo-KD can outperform all the relevant KD-based baseline methods. We have also added the experimental results in Appendix A.3 of the manuscript.
>
> *Table: Classification accuracy (\%) on DomainNet.*
>
> |Method|Clipart|Real|Sketch|Infograph|Painting|Quickdraw|Avg|
> |:-:|:-:|:-:|:-:|:-:|:-:|:-:|:-:|
> |Vanilla KD|50.6 ±0.2| 55.1 ±0.3 | 40.7 ±0.3 | 17.9 ±0.2 | 42.5 ±0.4 | 9.8 ±0.3  | 34.3 ±0.3 |
> |ANL-KD|52.5 ±0.4| 57.0 ±0.3 | 41.9 ±0.3 | 19.6 ±0.2 | 44.6 ±0.3 | 11.3 ±0.3 | 37.8 ±0.3 |
> |ADA-KD|53.6 ±0.2| 57.9 ±0.4 | 42.3 ±0.3 | 20.5 ±0.2 | 45.1 ±0.3 | 12.4 ±0.4 | 38.7 ±0.4 |
> |WLS-KD|53.1 ±0.4 | 57.7 ±0.3 | 42.8 ±0.2 | 21.1 ±0.1 | 45.3 ±0.2 | 12.0 ±0.2 | 38.5 ±0.3 |
> |RW-KD|51.3 ±0.3| 56.2 ±0.3 | 41.2 ±0.2 | 18.8 ±0.3 | 43.7 ±0.2 | 10.6 ±0.3 | 37.1 ±0.2 |
> |TGeo-KD|**55.2** ±0.3 | **59.3** ±0.4 | **44.6** ±0.1 | **21.7** ±0.3 | **46.9** ±0.3 | **14.1** ±0.3 | **40.2** ±0.3 |

---

> ### Author Response · Authors · 2023-11-19
>
> **Q3: Performance comparison with additional baselines.**
>
> **A3:** We conduct a comparative analysis between our proposed TGeo-KD and the two baselines (i.e., DKD [8] and SR-KD [9]) using CIFAR-100 and ImageNet, as illustrated in the tables below. The baseline results marked with * are obtained through our re-implementation of both student and teacher networks, and the average classification accuracy  (with standard deviation) are calculated over 5 repeated runs. For the remaining baseline results, we utilize the results provided or verified by these two baselines [8,9]. The best performance is bold.
>
> To demonstrate the significance of improvement, we conduct a statistical t-test, calculating t-scores calculated based on the top-1 classification accuracy of our TGeo-KD and the baselines. Although DKD [8] (with teacher: ResNet-32x4, student: ShuffleNetV2) and SR-KD [9] (with teacher: ResNet-50, student: MobileNetV1) showcase marginal improvements on CIFAR-100 and ImageNet, our computed t-scores consistently exceed the threshold value $t_{0.05,5}$ when employing the remaining network architectures. This result indicates the acceptance of the alternative hypothesis at a statistical significance level of 5.0%, and supports the conclusion that TGeo-KD consistently demonstrates a notable performance enhancement.
>
>
> *Table: Top-1 classification accuracy (\%) on CIFAR-100 dataset.*
> | Teacher| WRN-40-2 | ResNet-56 | ResNet-110 | ResNet-110 | ResNet-32x4 | ResNet-32x4 | ResNet-32x4 | WRN-40-2 |
> |:---------------:|:----------:|:-----------:|:------------:|:------------:|:-------------:|:-------------:|:-------------:|:----------:|
> | Student       | WRN-40-1 | ResNet-20 | ResNet-32  | ResNet-20  | ResNet-8x4  | ShuffleNetV1| ShuffleNetV2| ShuffleNetV1|
> | DKD           | 74.81    | 71.91     | 74.11      | 72.24*     | 76.32       | 76.45       | **77.07**       | 76.70    |
> | SR-KD         | 74.75    | 71.44     | 73.80      | 71.57      | 75.92       | 75.66       | 76.40       | 76.61    |
> | **TGeo-KD**       | **75.43**| **72.98** | **75.09**  | **73.55**  | **77.27**   | **76.83**   | 76.89   | **77.05**|
>
> *Table: Top-1 classification accuracy (\%) on ImageNet dataset.*
> | Network architecture | Teacher: ResNet-34; Student: ResNet-18 | Teacher: ResNet-34; Student: ResNet-18 | Teacher: ResNet-50; Student: MobileNetV1 |  Teacher: ResNet-50; Student: MobileNetV1     |
> |:-:|:-----------:|:---------------:|:----------------:|:--------------:|
> |Method|Top-1 ACC|Top-5 ACC|Top-1 ACC|Top-5 ACC|
> |DKD|71.70|   90.41   |71.87*|   90.09*  |
> |SR-KD|71.73|   90.60   |**72.49**|   90.92   |
> |**TGeo-KD**|**72.89**|   **91.80**   |72.46|   **90.95**   |
>
>
> **Q4: Definition of outliers (i.e., in ImageNet and CIFAR).**
>
> **A4:** In Section 4.4, we generate outliers by introducing synthetic Gaussian noise as additional training samples following the setting outlined in the studies [10,11,12]. In the context of Gaussian noise outliers, each RGB value for every pixel is sampled from an independent and identically distributed Gaussian distribution with a mean of 0.5 and unit variance, and each pixel value is clipped to the range $[0, 1]$.
>
> Ref:
> * [1] Pan et al. “Developing a Hybrid Intrusion Detection System Using Data Mining for Power Systems." Trans Smart Grid, 2015.
> * [2] Adhikari et al. “Applying Non-Nested Generalized Exemplars Classification for Cyber-Power Event and Intrusion Detection.” Trans Smart Grid, 2018.
> * [3] Hu et al. "Reinforcement Learning-Based Adaptive Feature Boosting for Smart Grid Intrusion Detection." Trans Smart Grid, 2023.
> * [4] Shan et al. "Deep Crossing: Web-Scale Modeling without Manually Crafted Combinatorial Features". KDD, 2016.
> * [5] Juan et al. "Field-aware Factorization Machines for CTR Prediction." RecSys, 2016.
> * [6] Zhu et al. "Ensembled CTR Prediction via Knowledge Distillation." CIKM, 2020.
> * [7] Peng et al. “Moment matching for multi-source domain adaptation.” ICCV, 2019.
> * [8] Zhao et al. “Decoupled knowledge distillation.” CVPR, 2022
> * [9] Yang et al. “Knowledge distillation via softmax regression representation learning.” ICLR, 2021.
> * [10] Hendrycks et al. “A baseline for detecting misclassified and out-of-distribution examples in neural networks.” ICLR, 2016.
> * [11] Liang et al. “Enhancing the reliability of out-of-distribution image detection in neural networks.” ICLR, 2018.
> * [12] Vyas et al. “Out-of-Distribution Detection Using an Ensemble of Self Supervised Leave-out Classifiers.” ECCV, 2018.

---

> > ### Author Response · Authors · 2023-11-22
> > **Kind Reminder to Reviewer**
> >
> > Dear reviewer,
> >
> > Thank you for your time and valuable feedback. We kindly remind you that the deadline for the Author/Reviewer Discussion is approaching. Have we adequately addressed your questions and concerns? We look forward to hearing from you soon.
> >
> > Best regards,
> >
> > Authors

---

### Official Review · Reviewer_Ur13 · 2023-11-03

**Soundness:** 2 fair
**Presentation:** 2 fair
**Contribution:** 2 fair
**Rating:** 5
**Confidence:** 4

**Summary:**

This paper discusses the concept of knowledge distillation (KD) and introduces a new method called TGeo-KD for determining sample-wise knowledge fusion ratios in KD. The traditional approach to determining fusion ratios in KD involves using a fixed value for all samples or gradually decreasing the ratio. The authors argue that considering the discrepancy between the student's and teacher's predictions (ST discrepancy) is crucial in determining the fusion ratio. They propose TGeo-KD, which leverages the trilateral geometry among the signals from the student, teacher, and ground truth to model the fusion process. Experiments conducted on CIFAR-100 dataset demonstrate the effectiveness of TGeo-KD compared to other methods. The main contributions of TGeo-KD include its use of trilateral geometry, mitigation of outliers, and superior performance across different tasks.

**Strengths:**

1.	The paper introduces a novel method called TGeo-KD for determining sample-wise knowledge fusion ratios in knowledge distillation. This method leverages trilateral geometry and captures the geometric relations between the signals from the student, teacher, and ground truth. By introducing this new method, the article contributes to the existing body of research in knowledge distillation.
2.	The article emphasizes that TGeo-KD is versatile and adaptable to different architectures and model sizes. This implies that the proposed method can be applied to a wide range of machine learning tasks and scenarios. The ability to apply TGeo-KD to different models and architectures enhances its practical utility and makes it a valuable contribution to the field of knowledge distillation.
3.	TGeo-KD leverages trilateral geometry at both intra-sample and inter-sample levels, which helps in mitigating the impact of outliers in the training samples. Outliers are samples that deviate significantly from the majority of the dataset and can negatively affect the learning process. By incorporating the trilateral geometry and considering the ST discrepancy, TGeo-KD provides a robust approach that is less sensitive to outliers, resulting in more stable and reliable knowledge transfer.

**Weaknesses:**

Complexity and Computational Cost: The TGeo-KD method proposed in the article leverages trilateral geometry and introduces a neural network to learn the fusion ratios. This suggests that the method may have a higher computational cost compared to simpler knowledge distillation techniques. Implementing and training the neural network for determining fusion ratios may require additional computational resources and time, which could be a potential drawback, especially in resource-constrained environments.

The proposed TGeo-KD leverages trilateral geometry and captures the geometric relations between the signals from the student, teacher, and ground truth. However, I think it lacks more sufficient theoretical proof.

**Questions:**

Please see the weakness.

---

> ### Author Response · Authors · 2023-11-19
>
> We appreciate your time and feedback. We would also appreciate your agreement on the novelty, effectiveness, robustness, and adaptability of our method. Our reply to your questions is as follows.
>
> **Q1: Complexity and computational cost.**
>
> **A1:** Thanks for the comments. We put the computational comparison in Appendix A.4 as shown in Table 15. In the experiment, we compare the computational cost of our proposed TGeo-KD with the baselines including vanilla KD [1], ANL-KD [2], ADA-KD [3], and WLS-KD [4]. We run experiments on CIFAR-100, and report the mean training time (in seconds) on a batch of 128 images per iteration. The student network is ResNet-32. The experiment is conducted on one NVIDIA RTX-3090 GPU. As shown in the table below (Table 15 in the manuscript), our method has little additional training time, making this method useful in practice.
>
> *Table: Mean training time (in seconds) on a batch for CIFAR-100 per iteration.*
> | Vanilla-KD | ANL-KD | ADA-KD | WLS-KD | TGeo-KD (Ours) |
> |:------------:|:--------:|:--------:|:--------:|:----------------:|
> | 0.095      | 0.095  | 0.101  | 0.097  | 0.097          |
>
> The reason why our method does not significantly increase processing time can be attributed to the additional neural network $f(\cdot)$ we employ, which is merely a 2-layer MLP as outlined in Section 4.5. Specifically,
> \begin{aligned}
> \alpha_i:=f_{\omega}(\Delta_i)=\text{sigmoid}\bigg(\mathbf{W}_2\cdot \Big(\text{relu}(\mathbf{W}_1\cdot \Delta_i +  \mathbf{b}_1)\Big) + \mathbf{b}_2\bigg),
> \end{aligned}
>
> where $\mathbf{W}_1\in\mathbb{R}^{h\times s}$, $\mathbf{W}_2\in\mathbb{R}^h$, $\mathbf{b}_1\in\mathbb{R}^h$, and $\mathbf{b}_2\in\mathbb{R}$ are the learnable parameters $\omega$, $h$ is the size of the hidden layer, and $s$ is the size of $\Delta_i$. In practice, we set $h\in\{16, 32, 64\}$, and $s=9*C$ (i.e., $C$ is the number of classes in the dataset) if $\Delta_i$ is defined as Eq. (10). Compared to the larger scale of the teacher and student models (e.g., ResNet-50 has 50 layers with over 25.6 million parameters), the relative size of this MLP is quite minimal. Consequently, the added time due to training this additional network is almost negligible.
>
> **Q2: Lack more sufficient theoretical proof.**
>
> **A2:** We appreciate your valuable feedback on the theoretical proof of our research. We would like to note that the key findings and contributions of this work revolve around empirical findings. Developing a theoretical proof for KD is challenging in general (even for the most influential works in the literature [1,2,5,6,7]), but could definitely be a direction for our future work. The only work we are aware of is [4], which presents a bias-variance perspective for KD. However, there are still several obstacles to adopting the results in [4] for TGeo-KD, as they do not account for the trilateral geometry in the data, which is the central element in our research. On the other hand, we note the recent work on the generalization bounds of triplet learning [8], which could be exploited to analyze trilateral geometry in TGeo-KD. Given the constraints of the limited rebuttal time, it is currently challenging for us to develop a thorough theoretical proof. We plan to address this as part of our future work.
>
>
> Ref:
> * [1] Hinton et al. “Distilling the knowledge in a neural network.” NeurIPS, 2014.
> * [2] Clark et al. “BAM! Born-Again Multi-Task Networks for Natural Language Understanding.” ACL, 2019.
> * [3] Lukasik et al. “Teacher’s pet: understanding and mitigating biases in distillation.” TMLR, 2022.
> * [4] Zhou et al. “Rethinking soft labels for knowledge distillation: A bias-variance tradeoff perspective.” ICLR, 2021.
> * [5] Li et al. “Curriculum Temperature for Knowledge Distillation.” AAAI, 2023.
> * [6] Beyer et al. “Knowledge distillation: A good teacher is patient and consistent.” CVPR, 2022.
> * [7] Huang et al. “Knowledge Distillation from A Stronger Teacher.” NeurIPS, 2022.
> * [8] Chen et al. “On the Stability and Generalization of Triplet Learning.” AAAI, 2023.

---

> > ### Author Response · Authors · 2023-11-22
> > **Kind Reminder to Reviewer**
> >
> > Dear reviewer,
> >
> > Thank you for your time and valuable feedback. We kindly remind you that the deadline for the Author/Reviewer Discussion is approaching. Have we adequately addressed your questions and concerns? We look forward to hearing from you soon.
> >
> > Best regards,
> >
> > Authors

---

> ### Comment · Reviewer_Ur13 · 2023-11-23
> **Re: Response**
>
> Thanks for authors' reponses to my questions. After reading the authors' responses and other reviewers' comments, some of my concerns have been addressed. However, I remain unconvinced by the theoretical aspect. In the Introduction section of the manuscript, the authors exclusively presented one experimental case to illustrate its core motivation. I anticipate encountering additional theoretical support for this motivation. It is crucial to ascertain whether the claim holds true when applied to a different dataset, as it currently lacks comprehensive validation. In addition, based on Table 1 and 2, it is evident that the proposed method yields slight gains, intensifying skepticism regarding the underlying motivation of this work.

---

> ### Author Response · Authors · 2023-11-23
>
> **Q1: Lack of sufficient theoretical support.**
>
> **A1:** We thank you for your continued engagement with the theoretical aspects of our research. In fact, we have been focusing on the theoretical development of our method. Eventually, we realized that by exploiting the notion of *performance gap* introduced by recent work [1], indeed, we can investigate the connections between teacher and ground truth, and how they affect student learning in our TGeo-KD.
>
> Specifically, given $\mathcal{D}$ and $f_\omega$, let $\theta_{\text{KD}}$ and $\theta_{\text{GT}}$, respectively, be the minimizers of $J_{\text{KD}}  \triangleq \frac{1}{N}\sum_{i=1}^N f_{\omega}(\Delta_i)\mathcal{L}^{\text{KD}}_i$ and
>
> $J_{\text{GT}} \triangleq\frac{1}{N}\sum_{i=1}^N (1-f_{\omega}(\Delta_i))\mathcal{L}^{\text{GT}}_i$.
>
> Then, we can define the *performance gap* for TGeo-KD as
> \begin{align}
>       \nabla = \nabla_{\text{KD}} + \nabla_{\text{GT}},
> \end{align}
>
> where $\nabla_{\text{KD}} = J_{\text{KD}}(\theta_{\text{GT}})-J_{\text{KD}}(\theta_{\text{KD}})$ and $J_{\text{GT}}(\theta_{\text{KD}})-J_{\text{GT}}(\theta_{\text{GT}})$.
> Note that:
>
> 1. $J_{\text{KD}}$ and $J_{\text{GT}}$ are both  *weighted* objective functions where the weights are determined  by $f_\omega$. In addition, $\theta_{\text{KD}}$ and $\theta_{\text{GT}}$ are the minimizers of  $J_{\text{KD}}$ and $J_{\text{GT}}$. Therefore, given the dataset $\mathcal{D}$, $\nabla$ is essentially a function of $f_\omega$.
> 2. Intuitively, if the labels predicted by the teacher model are quite different from the ground truth (i.e., a misleading teacher), one can expect that $\theta_{\text{KD}}$ and $\theta_{\text{GT}}$ will be quite different. As a result, by definition, $\theta_{\text{KD}}$ and $\theta_{\text{GT}}$ (hence $\nabla$) will be large. In other words, it essentially captures the relationship between the teacher's predictions and the ground truth.
>
> Let $\theta^*$ be the minimizer of the inner objective function of TGeo-KD (i.e., Eq. (5) in the manuscript). Then, given the definition of $\nabla$, we can prove that the generalization error $\mathbb{E}[\mathcal{L}(\theta^*)]$ can be upper bounded by
> \begin{align}
>        \mathbb{E}[\mathcal{L}(\theta^*)]  & \le \mathcal{J} (\theta^*;\omega)    + \mathcal{O}\left(\left(\frac{\frac{\gamma N}{2}+ \sum_{i=1}^N f_{\omega}^2(\Delta_i) + \sum_{i=1}^N (1- f_{\omega}(\Delta_i))^2}{ N^2} + \frac{\gamma C}{2N}  B(\omega)  \right)\sqrt{N}\right),
> \end{align}
>
> where we let ${N_{\text{train}}} =N$ to simplify the notation, and $f_{\omega}(\Delta_i) \le \gamma, \forall i = 1,\dots,N$ is the upper bound of $f_{\omega}(\Delta_i)$, $C$ is a constant with respect to TGeo-KD. Moreover, $B(\omega)$ can be upper bounded by
> \begin{align}
>        B(\omega) \le \mathcal{O}(\sqrt{\nabla + C'}),
> \end{align}
>
> where $C'$ is a constant with respect to TGeo-KD.
>
> From the theoretical results, we have the following observations:
>
> 1. $\Gamma  = \sum_{i=1}^N f_{\omega}^2(\Delta_i) + \sum_{i=1}^N (1- f_{\omega}(\Delta_i))^2$. Note that $\Gamma$ is maximized when $f_{\omega} \equiv 1 \text{ or } 0$. This in fact reflects a common sense in KD: one should not always trust the teacher or even the ground truth.
> 2. The upper bound of $B(\omega)$ reveals the importance of $\nabla$, and motivates how to control the tradeoff in trilateral relations. In particular, for a specific data point, when the teacher's prediction is wrong, one should obviously trust the ground truth more in order to obtain a better generalization performance. On the other hand, the teacher's prediction is correct but the student's and the teacher's predictions are quite different, its weight should be increased to reduce the performance gap (this can be seen by treating $\theta_{\text{KD}}$ and $\theta_{\text{GT}}$ as the surrogates of teacher and the ground truth), which is consistent with our empirical findings.
> 3. On the other hand, directly minimizing the upper bound is not straightforward (due to the definition of performance gap). In [1], this issue is sidestepped by a boosting-based algorithm, while we formulate it as a bilevel minimization problem.

---

> > ### Author Response · Authors · 2023-11-23
> >
> > In addition, we would also like to highlight the key differences between our work and [1]:
> >
> > 1. [1] proposes the notion of performance gap and study it in the contexts of transfer learning and multitask learning, while we study the KD problem. Note that while we exploit some techniques and concepts there (The scheme of proof of our theorem follows Remark 7 in [1]), the theoretical development is not trivial -- we need to define performance gap specifically for KD with different assumptions, and hence the generalization bound is also different.
> > 2. As we study KD rather than transfer or multitask learning in this paper, the proposed algorithm is also definitely different (e.g, exploiting the trilateral geometric relations, and solving it by bilevel optimization). More importantly, [1] is theoretically motivated, while our  TGeo-KD algorithm is empirically motivated by our findings shown in Figure 1.
> >
> > In fact, several existing works [2-5] explore the connections between transfer learning and KD from an algorithmic perspective. To the best of our knowledge, however, no one studied them from a theoretical aspect. As we mentioned earlier, the theoretical developments are missing even for the most influential works in KD. Our response here could be the first step towards this goal, and once again, we thank the reviewer for encouraging us to conduct the theoretical analysis.
> >
> > On the other hand, we have to admit that the current theoretical results are still premature. In particular, the generalization bound holds for any "fixed" weights $f_\omega$. In order words, it provides a high-level principle for learning the weights, but does not consider the learning dynamics of $f_\omega$ (i.e., the bilevel procedure). Similarly, the theory does not investigate the effect of $\Delta_i$ in TGeo-KD. We consider this an important direction for our future work. Lastly, we believe that you, as an expert in this field, understand that developing theoretical justifications for KD is challenging, especially within such a short period. We have tried our best to achieve this goal, and we hope that our response can address your concerns.

---

> > > ### Author Response · Authors · 2023-11-23
> > >
> > > **Q2: Slight performance gains on CIFAR-100 and ImageNet.**
> > >
> > > **A2:** Regarding the marginal improvements noted in Table 1 and Table 2, we would like to note the significance of these findings in specific scenarios and contexts within our field. While these improvements may appear modest, they are an important step forward in our research area.
> > > Among all the baselines, WLS-KD [6] is the best work with a marginal improvement of 0.27\% and 0.25\% over the second-best baseline ADA-KD [7] when utilizing ResNet-8x4 and ResNet-18 as the student model on CIFAR-100 and ImageNet, respectively. Employing the same student architecture as mentioned above, our TGeo-KD consistently demonstrates substantial performance enhancement, surpassing WLS-KD [6] (i.e., the best baseline), with improvements of 1.22\% and 1.10\%. Beyond this, our consistent performance improvements across various datasets in different domains demonstrate the broader applicability and generalization of our approach, not limited to a single dataset.
> > >
> > >
> > > Ref:
> > > * [1] Wang et al. "Gap Minimization for Knowledge Sharing and Transfer." JMLR, 2023.
> > > * [2] Liu et al. "Data-free knowledge transfer: A survey." arXiv preprint arXiv:2112.15278 (2021).
> > > * [3] Li et al. "Knowledge distillation with attention for deep transfer learning of convolutional networks." TKDD, 2021.
> > > * [4] Zhao et al. "Multi-source distilling domain adaptation." AAAI, 2020.
> > > * [5] Ao et al. "Fast generalized distillation for semi-supervised domain adaptation." AAAI, 2017.
> > > * [6] Zhou et al. “Rethinking soft labels for knowledge distillation: A bias-variance tradeoff perspective.” ICLR, 2021.
> > > * [7] Lukasik et al. “Teacher’s pet: understanding and mitigating biases in distillation.” TMLR, 2022.

---

### Comment · Area_Chair_MFzQ · 2023-11-21
**[Time Sensitive, ICLR24] Please read authors' responses and try to discuss the remaining concerns with the authors**

Dear Reviewers,

The authors have provided detailed responses to your comments.

Could you have a look and try to discuss the remaining concerns with the authors? The reviewer-author discussion will end in two days.

We do hope the reviewer-author discussion can be effective in clarifying unnecessary misunderstandings between reviewers and the authors.

Best regards,

Your AC

---

### Meta-Review · Area_Chair_MFzQ · 2023-12-05

**Metareview:**

This paper is the first to consider the effects of the discrepancy between the student’s prediction and the teacher’s prediction in KD. The motivation is very clear, and such discrepancy should be considered in determining the alpha value in KD. Then, this paper proposes a corresponding method that incorporates the discrepancy in determining the alpha. The experiments are solid to verify the claim made in this paper. However, more theoretical analysis should be included in this paper to further strengthen the contribution of this novel idea.

**Justification For Why Not Higher Score:**

Although the contribution is solid, it is mainly based on experiments instead of theoretical analysis. This limits the contribution of this paper in general.

**Justification For Why Not Lower Score:**

This paper is the first to consider the effects of the discrepancy between the student’s prediction and the teacher’s prediction in KD. The motivation is very clear, and such discrepancy should be considered in determining the alpha value in KD. This finding might motivate more follow-up ideas in KD. Thus, this paper can be accepted by ICLR.

---

### Decision · Program_Chairs · 2024-01-16

Accept (poster)